**DOI: 10.1038/ncomms13076**　　**OPEN**

# Unveiling the pentagonal nature of perfectly aligned single-and double-strand Si nano-ribbons on Ag(110)

Jorge I. Cerdá[1], Jagoda Sławińska[1], Guy Le Lay[2], Antonela C. Marele[3], José M. Gómez-Rodríguez[3,4,5] & María E. Dávila[1]

Carbon and silicon pentagonal low-dimensional structures attract a great interest as they may lead to new exotic phenomena such as topologically protected phases or increased spin–orbit effects. However, no pure pentagonal phase has yet been realized for any of them. Here we unveil through extensive density functional theory calculations and scanning tunnelling microscope simulations, confronted to key experimental facts, the hidden pentagonal nature of single- and double-strand chiral Si nano-ribbons perfectly aligned on Ag(110) surfaces whose structure has remained elusive for over a decade. Our study reveals an unprecedented one-dimensional Si atomic arrangement solely comprising almost perfect alternating pentagons residing in the missing row troughs of the reconstructed surface. We additionally characterize the precursor structure of the nano-ribbons, which consists of a Si cluster (nano-dot) occupying a silver di-vacancy in a quasi-hexagonal configuration. The system thus materializes a paradigmatic shift from a silicene-like packing to a pentagonal one.

[1] Instituto de Ciencia de Materiales de Madrid, ICMM-CSIC, Cantoblanco, 28049 Madrid, Spain. [2] Aix Marseille Université, CNRS, PIIM UMR 7345, 13397 Marseille, France. [3] Departamento de Física de la Materia Condensada, Universidad Autónoma de Madrid, E-28049 Madrid, Spain. [4] Condensed Matter Physics Center (IFIMAC), Universidad Autónoma de Madrid, E-28049 Madrid, Spain. [5] Instituto Nicolás Cabrera, Universidad Autónoma de Madrid, E-28049 Madrid, Spain. Correspondence and requests for materials should be addressed to J.I.C. (email: jcerda@icmm.csic.es) or to G.L.L. (email: guy.lelay@univ-amu.fr).

From the simplest cyclopentane ring and its numerous organic derivates to their common appearance in extended geometries such as edges or defects in graphene, pentagons are frequently encountered motifs in carbon-related systems. Even a penta-graphene Cairo-type two-dimensional (2D) structure has been proposed as a purely pentagonal C allotrope with outstanding properties competing with those of graphene[1]. Conversely, pentagonal Si motifs are hardly found in nature. Despite the large effort devoted to design Si-based structures analogous to those of carbon, the existence of Si pentagonal rings has only been reported in clathrate bulk phases[2] or in complex Si reconstructions[3,4]. Several theoretical studies have hypothesized stable Si pentagonal structures either in the form of one-dimensional (1D) nanotubes[5,6] or at the reconstructed edges of silicene nano-ribbons (NRs)[7,8] or even as hydrogenated penta-silicene[9] or highly corrugated fivefold coordinated siliconeet[10] 2D sheets, the latter recognized as a topological insulator[11]. However, to date none of them have yet been synthesized.

In the present work we unveil, via extended density functional theory (DFT)[12] calculations and scanning tunnelling microscopy (STM) simulations[13,14], the atomic structure of 1D Si NRs grown on the Ag(110) surface. Our analysis reveals that this system constitutes the first experimental evidence of a silicon phase solely comprising pentagonal rings. The possibilities that this unprecedented 1D topography opens are manifold, ranging from Si-based nano-wires in circuits, enlarged spin–orbit effects or even the realization of a new Si allotrope.

## Results

**STM and X-ray Photoemission Spectroscopy (XPS)**. Since their discovery in 2005 (ref. 15) the atomic structure of Si NRs on Ag(110) has remained elusive and strongly disputed[15–23]. Figure 1 presents a summary of Si NRs measured with STM. The structures were obtained after Si sublimation onto a clean Ag(110) surface at room temperature. Figure 1a,c corresponds to a low Si coverage image with an isolated nano-dot structure and a single-strand NR (SNR) 0.8 nm wide running along the [110] direction with a 2× periodicity. The SNR topography consists of alternating protrusions at each side of the strand with a glide plane. At higher coverages and after a mild annealing, a dense and highly ordered phase is formed (Fig. 1e) consisting of double-strand NRs (DNRs) with a 5× periodicity along the [001] direction again exhibiting a glide plane along the centre of each DNR. The images are in perfect accord with previous works[15,17,22,23]. Further key information on the system is provided by the high-resolution Si-2p core level photoemission spectrum for the DNRs displayed in Fig. 1g—that for the SNRs is almost identical[24]. The spectrum can be accurately fitted with only two (spin–orbit splitted) components having an intensity ratio of roughly 2:1 (we estimate a maximum error of 20% in the $Si_s$:$Si_{ad}$ intensity ratio based on analogous spectra recorded at different energies or even beamlines[25]). Furthermore, previous angular resolved photoemission (ARPES) experiments[25] assigned the larger and smaller components to subsurface $Si_s$ and surface $Si_{ad}$ atoms, respectively, indicating that the NRs comprise two different types of Si atoms, with twice as many $Si_s$ as $Si_{ad}$.

**Nano-dot's quasi-hexagonal structure**. We first focus on the nano-dot shown in Fig. 1a, as it may be regarded as the precursor structure for the formation of the extended NRs. The nano-dot exhibits a local *pmm* symmetry with two bright protrusions aligned along the [001] direction, each of them having two adjacent dimmer features along the [110] direction. After

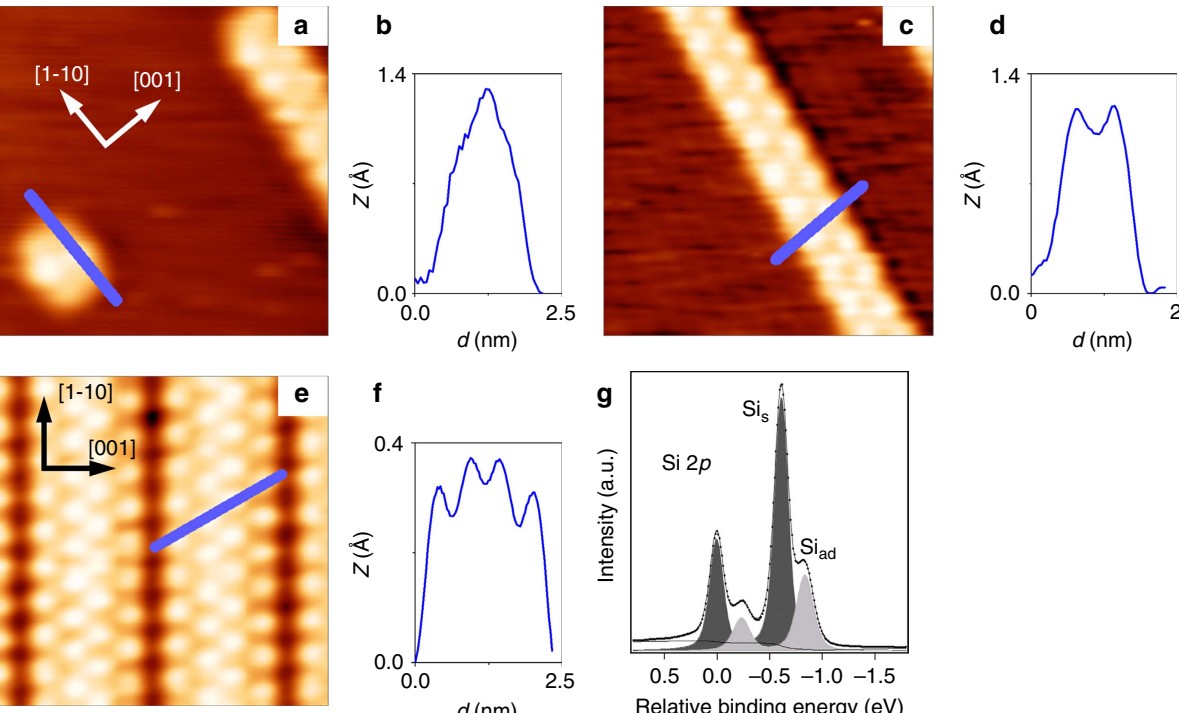

**Figure 1 | Experimental results based on STM and X-ray Photoemission Spectroscopy techniques.** (**a**,**c**,**e**) 5.3 × 5.3 nm$^2$ STM images of the Si nanostructures on Ag(110) studied in this work: (**a**) a Si nano-dot; (**c**) a Si SNR; and (**e**) an extended Si DNR phase. (**b**,**d**,**f**) Profiles along the solid lines passing over the maxima in the STM images. The tunnelling parameters are (**a**,**b**) −1.5 V, 2.4 nA, (**b**,**d**) −1.8 V, 1.2 nA and (**e**,**f**) 1.3 V, 1.1 nA. (**g**) Deconvolution of the Si-2p core level photoemission spectra recorded at normal emission and at 135.8 eV photon energy for the Si DNRs structure (original spectra adapted from ref. 24).

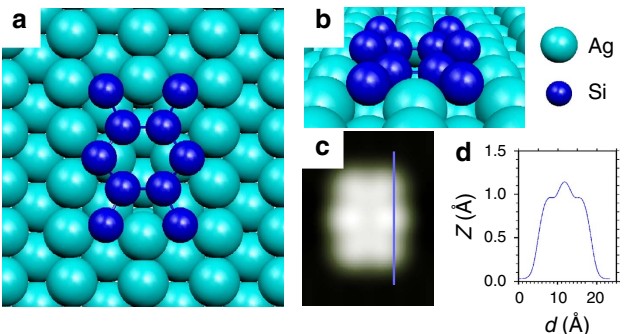

**Figure 2 | Nano-dot model structure and STM simulations.** (**a**,**b**) Top and perspective views of the optimized low-coverage nano-dot structure comprising 10 Si atoms located in a Ag di-vacancy and following a quasi-hexagonal arrangement. (**c**) $(18 \times 24)$ Å$^2$ simulated STM topographic image for this structure and, (**d**) line profile along the blue solid line in **c**.

considering a large variety of trial models (Supplementary Fig. 1) we found that only one, shown in Fig. 2, correctly reproduces the experimental image both in terms of aspect and overall corrugation. It consists of a 10-atom Si cluster located in a double silver vacancy generated by removing two adjacent top row silver atoms. There are four symmetry-equivalent $Si_s$ atoms residing deeper in the vacancy, two $Si_{ad}^1$ in the middle, which lean towards short silver bridge sites and four outer $Si_{ad}^2$ residing at long bridge sites. The formers lie 0.8 Å above the top Ag atoms and are not resolved in the STM image, while the $Si_{ad}^1$ and $Si_{ad}^2$ protrude out of the surface by 1.4 and 1.1 Å thus leading to the six bump structure in the simulated image with the $Si_{ad}^1$ at the centre appearing brighter. Therefore, although the nano-dot shows marked differences with respect to the extended NRs, its structure already accounts for the presence of two distinct types of Si atoms at the surface ($Si_s$ and $Si_{ad}$) and confirms the tendency of the Ag(110) surface to remove top row silver atoms on Si adsorption[22,26], as could be expected from the low stability of this particular surface[27,28].

**Single-strand Si pentagonal NRs.** Inspired by the nano-dot Ag di-vacancy structure and by recent STM and grazing incidence X-ray diffraction measurements[26] pointing towards the existence of a missing row (MR) reconstruction along the [1$\bar{1}$0] direction of the Ag surface, we considered several trial structures for the SNRs by placing Si atoms in the MR troughs ($Si_s$) and next adding further adatoms ($Si_{ad}$) on top, while maintaining a 2:1 concentration ratio between the two. Figure 3a,b shows top and side views of the optimized geometry for the SNRs after testing several trial models (Supplementary Fig. 2 and Supplementary Table 1). It involves a MR and six Si atoms per cell. The new paradigm is the arrangement of the Si atoms into pentagonal rings running along the MR and alternating their orientation (we denote it as the pentagonal missing row (P-MR) model). Despite no symmetry restriction was imposed, the relaxed P-MR SNR belongs to the *cmm* group presenting two mirror planes plus an additional glide plane along the MR troughs (see Supplementary Fig. 3 for a detailed description). Apart from a considerable buckling of 0.7 Å between the lower Si atoms residing in the MR troughs ($Si_s$) and the higher ones ($Si_{ad}$) leaning towards short bridge sites at the top silver row, the pentagonal ring may be considered as rather perfect, with a very small dispersion in the Si–Si distances (2.35–2.37 Å) and bond angles ranging between 92° and 117°, that is, all close to the 108° in a regular pentagon. The associated STM image and line profile, Fig. 3c,d, show (symmetry) equivalent protrusions 1.3 Å high

at each side of the strand, in perfect agreement with the experimental image Fig. 1c. Still, since different models may yield similar STM images, a more conclusive gauge to discriminate among them is to examine their relative formation energies. In this respect, the energetic stability of the P-MR structure is far better ($\sim 0.1$ eV per Si) than all other SNR models considered (Supplementary Fig. 4 and Supplementary Table 2).

**Double-strand Si pentagonal NRs.** Within the pentagonal model the DNR structure may be naturally generated by placing two SNRs within a $c(10 \times 2)$ cell. However, since the P-MR SNRs are chiral, adjacent pentagonal rings may be placed with the same or with different handedness, leading to two possible arrangements among the enantiomers. Figure 3f–i displays the optimized geometry and simulated STM topography for the most stable (by 0.03 eV per Si) P-MR DNR configuration. The pentagonal structure in each NR is essentially preserved, the main difference with respect to the SNRs being the loss of the glide plane along the MR troughs replaced by a new one along the top silver row between adjacent SNRs. There is a slight repulsion between the NRs, which shifts them away from each other by around 0.2 Å. As a result, the $Si_{ad}$ at the outer edges of the DNR end up lying 0.07 Å higher than the inner ones making the alternating pentagons along each strand not strictly equivalent anymore. In the simulated STM image the outer maxima appear dimmer than the inner ones by 0.1 Å, which adopt a zig-zag aspect. The inversion in their relative corrugations is due to the proximity between the inner Si adatoms ($\sim 4$ Å) compared with the almost 6 Å distance between the inner and outer ones, so that the bumps of the formers overlap and lead to brighter maxima. All these features are in accordance with the experimental profiles shown in Fig. 1f. In fact, the P-MR DNR structure is the most stable among all other NR models considered for a wide range of Si chemical potentials ranging from Si-poor to -rich conditions (Supplementary Fig. 4).

**Electronic properties.** Figure 4 presents a summary of the electronic properties of the P-MR structure. Figure 4a shows an isosurface of the total electronic density for the SNRs. The $Si_s$ atoms in the pentagonal rings are clearly linked through an $sp^2$ type bonding (three bonds each) while the $Si_{ad}$, due to the buckling, show a distorted $sp^3$ type tetrahedral arrangement making bonds with two $Si_s$, as well as with the adjacent short bridge silver atoms in the top row. Figure 4b displays ARPES spectra for the SNR and DNR phases. Both energy distribution curves reveal Si-related peaks previously attributed to quantum well states originating from the lateral confinement within the NRs. For the SNRs three states are observed at $-1.0$, $-2.4$ and $-3.1$ eV binding energy, while for the DNRs one further peak is identified at $-1.4$ eV. The computed (semi-infinite) surface band structures projected on the Si pentagons (blue) and the silver MR surface (red) are superimposed in Fig. 4c,d for the SNRs and DNRs, respectively. Overall, within the expected DFT accuracy and experimental resolution, the maps satisfactorily reproduce the experimental spectra. At $\Gamma$ the SNRs present two sharp intense Si bands below the Fermi level (S1 and S3) and faint (broader) features arising from two almost degenerate bands (S4 and S5) and a dimmer state (S2). As expected, they are almost flat along $\bar{\Gamma} - \bar{X}$ while along $\bar{\Gamma} - \bar{Y}$ they present an appreciable dispersion and finally merge into two degenerate states at the high symmetry Y point. The orbital character of the S2–S5 bands is mainly $p_{xy}$ and may thus be assigned to localized $sp^2$ planar bonds. Conversely, band S1 is fully dominated by the $Si_s$-$pz$ states ($\pi$-band) and shows a strong downward dispersion along $\bar{\Gamma} - \bar{Y}$ due to hybridization with the metal $sp$ bands. Similarly, faint

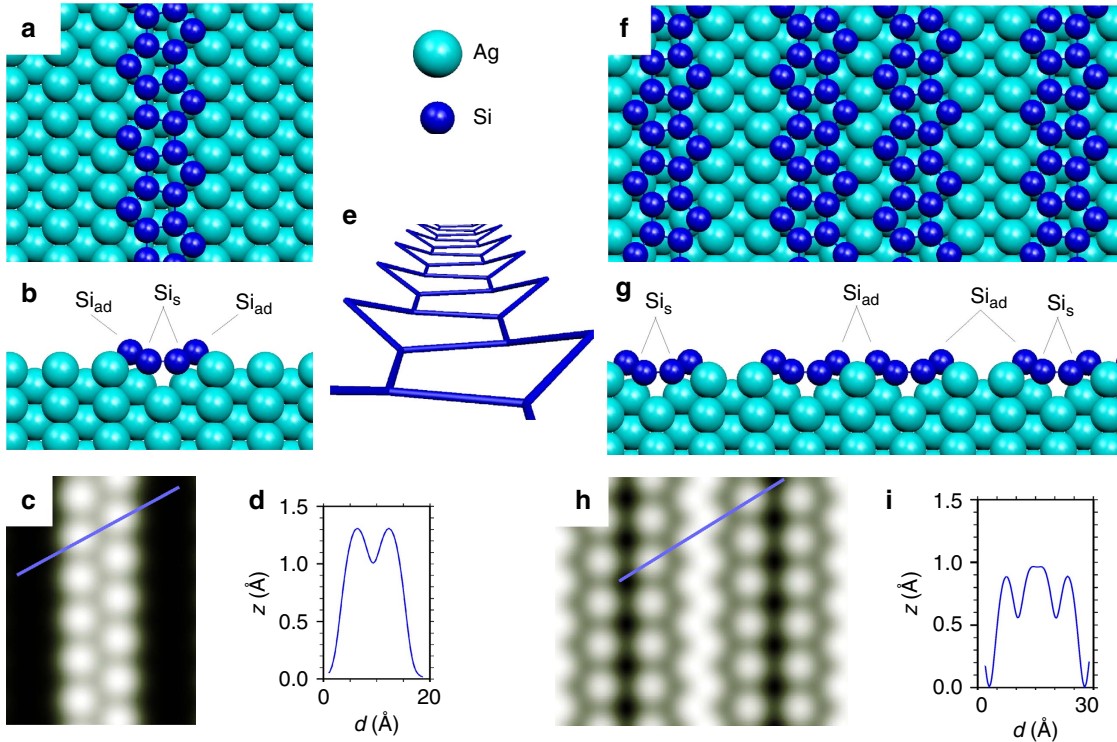

**Figure 3 | P-MR model for SNR and DNR and STM simulations.** Optimized geometries for the P-MR model. (**a,b**) Top and side views of the lower-coverage SNR phase, and (**c,d**) the simulated topographic STM image and line profile along the solid blue line. (**e**) Perspective view of a Si pentagonal strand without the silver surface. The strand is made of alternating buckled pentagons. (**f,g**) Top and side views of the high-coverage DNR phase and (**h,i**) the simulated topographic STM image and line profile. All STM simulations used a sharp Si ended tip apex (Supplementary Fig. 5) and set points $V = -0.2$ V and $I = 1$ nA.

dispersive bands of mainly $p_z$ character hybridizing with the metal appear in the empty state region. The electronic structure for the DNRs is similar to that of the SNRs, except that the number of Si bands is doubled and most of them become splitted and shifted due to the interaction between adjacent SNRs. Noteworthy is the appearance of an electron pocket at $\bar{\Gamma}$ associated to a parabolic Si-$p_z$ band with onset at $-0.5$ eV, as well as the large Fermi velocities ($\vec{v}_F \sim 10^6$ m s$^{-1}$) found in the linear part of the intense bands emerging from $\bar{\Gamma}$ (indicated by circles in Fig. 4d).

## Discussion

We have solved the long debated structure of silicon NRs on Ag(110), finding an unprecedented 1D Si pentagonal phase, which consists of adjacent inverted pentagons stabilized within the MR troughs. The model is in accordance with most of previous experimental results for this system: it involves a MR reconstruction as deduced from X-ray diffraction[26], comprises two types of Si atoms with a ratio 2:1 between the $Si_s$ and $Si_{ad}$ concentrations as seen by photoemission, accurately matches the STM topographs also explaining dislocation defects between NRs (Supplementary Fig. 6) and accounts for the quantum well states measured by ARPES. We have also determined the quasi-hexagonal geometry of a Si nano-dot inside a silver di-vacancy. This precursor structure for the NRs can be considered as the limiting process for expelling surface Ag atoms to create a MR along which the Si pentagons can develop. At this point, however, we cannot determine the precise diffusion mechanism or that behind the hexagonal-to-pentagonal transition (Supplementary Fig. 7). The discovery of this unique Si-based pentagonal phase puts hope on the realization of a new

1D Si allotrope, which could be achieved by weakening the NR–substrate interaction. Possible routes to this end could be intercalation of more inert species, their growth on a different substrate or hydrogenation processes. In fact, a pristine (free-standing) Si pentagonal strand is found to be metastable and when hydrogenated it even becomes more stable than when adsorbed on the Ag(110) surface (see Supplementary Fig. 8 for a summary of the atomic and electronic structure of freestanding pristine and hydrogenated Si pentagonal NRs). We are also convinced that our study will promote the synthesis of analogous exotic Si phases on alternative templates with promising properties[10].

## Methods

**Experimental.** For both types of prepared structures (isolated Si SNRs or ordered DNRs), the same procedure has been used for sample preparation: that is, the Ag(110) substrate was cleaned in the ultra-high vacuum chambers (base pressure: $9 \times 10^{-11}$ mbar) by repeated sputtering of Ar$^+$ ions and subsequent annealing of the substrate at 750 K, while keeping the pressure below $3 \times 10^{-10}$ mbar during heating. Si was evaporated at a rate of 0.03 ml min$^{-1}$ from a silicon source to form the NRs. The Ag substrate was kept at room temperature to form the isolated SNR 0.8 nm wide, while a mild heating of the Ag substrate at 443 K allows the formation of an ordered grating DNR 1.6 nm wide[24].

STM measurements were done with a home-made variable temperature ultra-high vacuum STM[29]. All STM data were measured and processed with the WSxM software[30]. High-resolution photoelectron spectroscopy experiments of the shallow Si-2p core levels and of the valence states, were carried out to probe, comparatively, the structure and the electronic properties of those nanostructures. The ARPES experiments were carried out at the I511 beamline of the Swedish Synchrotron Facility MAX-LAB in Sweden. The end station is equipped with a Scienta R4000 electron spectrometer rotatable around the propagation direction of the synchrotron light. It also houses low-energy electron diffraction and sputter cleaning set-ups. Further details on the beam line are given in ref. 31. In all the photoemission spectra the binding energy is referenced to the Fermi level. The total experimental resolution for core level and valence band spectra were 30 meV ($h\nu = 135.8$ eV for Si-2p) and 20 meV ($h\nu = 75$ eV for the valence band),

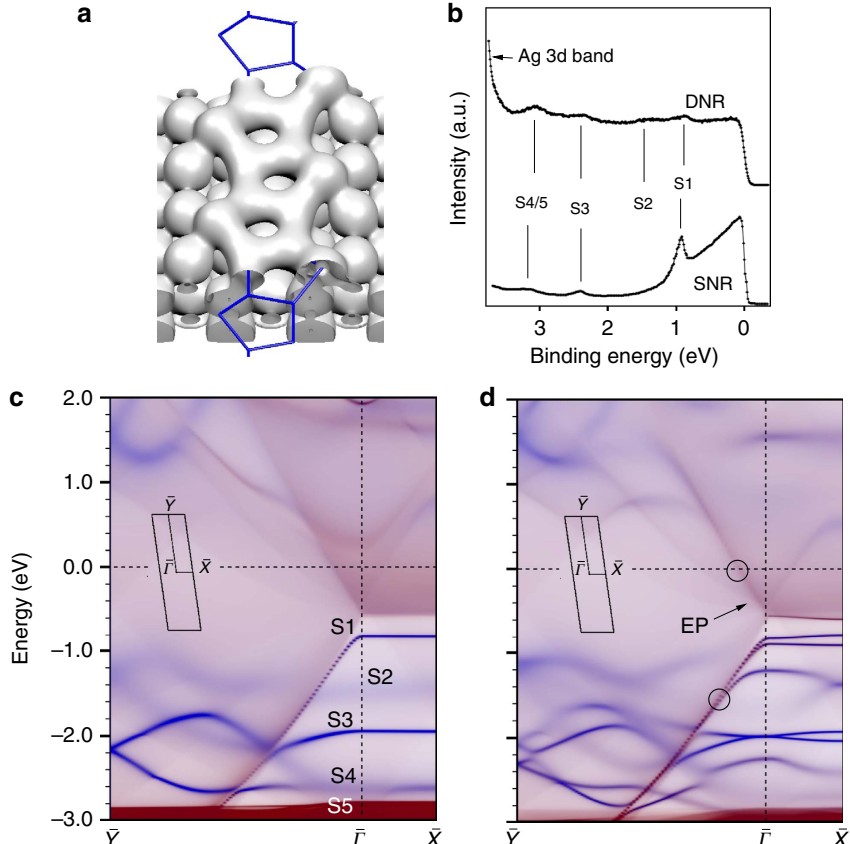

**Figure 4 | Electronic structure of the P-MR model. (a)** Charge density isosurface for the SNRs with blue sticks indicating the Si pentagons. **(b)** Energy distribution curves around the $\bar{X}$ point for the SNRs acquired at 78 eV photon energy (adapted from ref. 38) and for the DNRs at 75 eV (adapted from ref. 24). **(c,d)** Partial Density Of States (PDOS) $(\vec{k}, E)$ projected on the Si (blue) and Ag (red) atoms along the $\bar{Y} - \bar{\Gamma} - \bar{X}$ **k**-path (see insets) for the SNRs and DNRs, respectively. In **d** the Fermi velocities calculated at the location of the open circles are $\mathbf{v}_F = 1.3 \times 10^6 \, \mathrm{m \, s^{-1}}$ for the upper band and $\mathbf{v}_F = 1.0 \times 10^6 \, \mathrm{m \, s^{-1}}$ for the lower one.

respectively. A least-square fitting procedure was used to analyse the core levels, with two doublets, each with a spin–orbit splitting of $610 \pm 5$ meV and a branching ratio of 0.42. The Si-2$p$ core level collected at normal emission is dominated by the $\mathrm{Si_s}$ component. Its full width at half maximum is only 68 meV while the energy difference between the two $\mathrm{Si_s}$ and $\mathrm{Si_{ad}}$ components is 0.22 eV.

**Theory.** All calculations have been carried out at the *ab initio* level within the DFT using the SIESTA-GREEN package[12,13]. For the exchange–correlation interaction we considered both the local density[32] (LDA) as well as the generalized gradient[33] (GGA) approximations. Test calculations showed that including van der Waals corrections[34,35] yielded negligible changes in the optimized geometries and, therefore, they have been neglected (the same conclusion is reached in ref. 22). The atomic orbital basis set consisted of double-zeta polarized numerical orbitals strictly localized after setting a confinement energy of 100 meV in the basis set generation process. Real space three-centre integrals were computed over three-dimensional grids with a resolution equivalent to a 700 Rydbergs mesh cutoff. Brillouin zone integration was performed over *k*-supercells of around $(20 \times 28)$ relative to the Ag-$(1 \times 1)$ lattice while the temperature $kT$ in the Fermi–Dirac distribution was set to 100 meV.

All considered Si-NR-Ag(110) structures were relaxed using 2D periodic slabs involving nine metal layers with the NR adsorbed at the upper side of the slab. A c$(10 \times 2)$ supercell was used for both the SNR and DNR structures. In all cases, the Si atoms and the first three metallic layers were allowed to relax until forces were below $0.02 \, \mathrm{eV \, \mathring{A}^{-1}}$ while the rest of silver layers were held fixed to their bulk positions (for which we used our LDA (GGA)-optimized lattice constant of 4.07 Å (4.15 Å), slightly smaller (larger) than the 4.09 Å experimental value). For the nano-dot calculations, and given that a larger unit cell is required to simulate its isolated geometry, the atomic relaxations of all the trial models (Supplementary Fig. 1) were carried out for $(4 \times 5)$ or $(4 \times 6)$ supercells. Once the correct structure was identified (Supplementary Fig. 1), we further optimized it increasing the unit cell to a $(6 \times 10)$ to remove any overlaps between image cells (see Fig. 2 in the main text). Finally, for the penta-silicene free-standing calculations shown in Supplementary Fig. 8 we considered 1D strand geometries with a $\times 2$ periodicity with respect to the Ag(110) lattice parameter

along the $[1\bar{1}0]$ direction. The calculated stress in the strands was nevertheless small ($\sim 3 \times 10^{-3} \, \mathrm{eV \, \mathring{A}^{-3}}$).

**Band structure.** To examine the surface band dispersion we computed **k**-resolved surface projected density of states PDOS($\vec{\mathbf{k}}, E$) maps in a semi-infinite geometry. To this end we stacked the Si-NR and first metallic layers on top of an Ag(110) bulk-like semi-infinite block via Green's functions matching techniques following the prescription detailed elsewhere[14,36]. For this step we recomputed the slab's Hamiltonian using highly extended orbitals (confinement energy of just 10–20 meV) for the Si and Ag surface atoms in the top two layers (this way the spatial extension of the electronic density in the vacuum region is largely extended and the calculation becomes more accurate).

**STM simulations.** For the STM simulations we modelled the tip as an Ag(111) semi-infinite block with a one-atom-terminated pyramid made of 10 Si atoms stacked below acting as the apex (Supplementary Fig. 5). Test calculations using other tips (for example, clean Ag or clean W) did not yield any significant changes. Highly extended orbitals were again used to describe both the surface and the apex atoms thus reproducing better the expected exponential decay of the current with the tip-sample normal distance $z_{\mathrm{tip}}$. Tip-sample atomic orbital interactions were computed at the DFT level using a slab including the Si NR on top of three silver layers, as well as the Si tip apex. The interactions (Hamiltonian matrix elements) were stored for different relative tip-surface positions and next fitted to obtain Slater–Koster parameters that allow a fast and accurate evaluation of these interactions for any tip-sample relative position[14]. Our Green's function-based formalism to simulate STM images includes only the elastic contribution to the current and assumes just one single tunnelling process across the STM interface; it has been extensively described in previous works[13,14]. Here we used an imaginary part of the energy of 20 meV, which also corresponds to the resolution used in the energy grid when integrating the transmission coefficient over the bias window. We further assumed the so-called wide band limit at the tip[14] to alleviate the computational cost and remove undesired tip electronic features. The images were computed at different

biases between $-2$ and $+2$ V scanning the entire unit cell with a lateral resolution of 0.4 Å always assuming a fixed current of 1 nA. Nevertheless, the aspect of the images hardly changed with the bias, in accordance with most experimental results.

**Energetics.** To establish the energetic hierarchy among different Si NR structures we first computed their adsorption energies (per Si atom), $E_{ads}$, via the simple expression:

$$E_{ads} = (E_{tot}(N_{Ag}, N_{Si}) - E_{surf}(N_{Ag}) - N_{Si}E_{Si}^0)/N_{Si} \qquad (1)$$

where $N_{Si/Ag}$ are the number of Si and Ag atoms in the slab containing the NR and the Ag(110) surface, $E_{tot}(N_{Ag}, N_{Si})$ refers to its total energy, $E_{surf}(N_{Ag})$ the energy of the clean Ag surface without the NRs (but including any MRs) and $E_{Si}^0$ the energy of an isolated Si atom. In the low temperature limit equation (1) allows to discriminate between structures with the same number of silver and Si atoms (Supplementary Table 2).

However, a more correct approach to compare the NR's stabilities between structures with different Si and Ag concentrations is to compute their formation energies, $\gamma$, as a function of the Si and Ag chemical potentials, $\mu_{Si/Ag}$. To this end, we use the standard low temperature expression for the grand-canonical thermodynamic potential[37]:

$$\Omega(\mu_{Si}, \mu_{Ag}) = E_{tot}(N_{Si}, N_{Ag}) - N_{Si}\mu_{Si} - N_{Ag}\mu_{Ag} \qquad (2)$$

The chemical potentials may be obtained via $\mu_{Ag/Si} = E^{ref} - E_{Si}^0$, where $E_{Si}^0$ corresponds to the total energy of the isolated atom and $E^{ref}$ to that of a reference structure acting as a reservoir of Ag or Si atoms. Here we use the bulk f.c.c. phase for silver ($\mu_{Ag}^{LDA} = -4.67\,eV$ and $\mu_{Ag}^{GGA} = -3.60\,eV$), while that of Si is considered as a parameter (see below). The NR's formation energy, normalized to the Ag(110)-$(1 \times 1)$ surface unit cell area, then takes the form

$$\gamma = \frac{1}{N}\left[E_{tot}(N_{Si}, N_{Ag}) - N_{Ag}E_{Ag}^{ref} - N_{Si}\mu_{Si}\right] - \gamma_{Ag}^{sb} \qquad (3)$$

with $N = 10$ because the same c$(10 \times 2)$ was used for all NR structures and $\gamma_{Ag}^{sb}$ accounts for the formation energy of the unrelaxed surface at the bottom of the slab, which was obtained according to $\gamma_{Ag}^{sb} = \frac{1}{2}\left[E_{1 \times 1}(N_{Ag}) - N_{Ag}E_{Ag}^{ref}\right]$ with $E_{1 \times 1}(N_{Ag})$ giving the total energy of an unrelaxed nine layer-thick Ag(110)-$(1 \times 1)$ slab.

We follow the standard procedure of treating the Si chemical potential as a parameter in equation (3) and plot the formation energies for each structure as a function of $\mu_{Si}$ in Supplementary Fig. 4a,b for the LDA- and GGA-derived energies, respectively. However, since a reference structure for the Si reservoir is not available (and hence the absolute value of $\mu_{Si}$ is unknown) we plot the formation energies as a function of a chemical potential shift, $\Delta\mu_{Si}$, whose origin is placed at the first crossing between the formation energy of the clean Ag(110) and that of any of the NRs (in our case it corresponds to the P-MR DNR structure). Within this somewhat arbitrary choice, small or negative values of $\Delta\mu_{Si}$ would correspond to Si-poor conditions, while large positive values to Si-rich conditions.

Finally, the NR–H interaction strengths for the hydrogenated $P + n_H$ NRs shown in the Supplementary Fig. 8 were determined from the total energy of the free-standing pentagonal strand, $E_{NR}^0$, via

$$E^I(NR, n_H) = (-E(NR, n_H) + E_{NR}^0 + n_H E_H^0)/N_{Si} \qquad (4)$$

where $N_{Si} = 6$ and $n_H = 2, 4$ and $6$ are the total number of Si and H atoms per $\times 2$ cell and $E_H^0$ the total energy of an isolated H atom.

**Data availability.** The data that support the findings of this study are available from the corresponding author on request.

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

## Acknowledgements

This work has been funded by the Spanish MINECO under contract Nos. MAT2013-47878-C2-R, MAT2015-66888-C3-1R, CSD2010-00024, MAT2013-41636-P, AYA2012-39832-C02-01 and ESP2015-67842-P.

## Author contributions

J.I.C. and J.S. performed all the theoretical calculations; A.C.M., M.E.D. and J.M.G.-R. performed all the STM experiments; M.E.D. and G.L.L. performed the ARPES measurements; J.I.C. and M.E.D. conceived most of the novel model structures tested; J.I.C. and G.L.L. wrote the manuscript. All authors contributed to the manuscript and figure preparation.

## Additional information

**Competing financial interests:** The authors declare no competing financial interests.

