## [Peer Review File · Nature Communications]

Reviewers' comments:

Reviewer #1 (Remarks to the Author):

In this paper, the authors have re-checked the geometric structure of silicene on the Ag surface. Through DFT calculations and STM comparisons, they have found that unlike the previously proposed Si hexagons, the narrow Si nanoribbons on Ag(110) surface will be pentagonal. With the help of missing Ag rows, Si atoms are localized in these troughs and form "penta-silicene" nanoribbons on the surface. In fact, such formation of pentagonal structures is similar to those in nanotubes, which is a natural result of the space constraint. The formed pentagonal Si nanoribbons also exhibit linear dispersions just below the Fermi level, which is in line with the previous experiments. The present paper gives a new interpretation for the grown Si nanostructure on Ag surface, which is helpful to understand their novel behaviors. Thus, I recommend the publication of this paper on the Nature Communications.

Some additional queries are as follows:

- (1) Since the isolated Si nano-dot still keeps the hexagonal structure. How do they evolve to the pentagonal structure for extended stripes? If the initial Si₁₀ atoms are placed in a trough (not just in an Ag di-vacancy), would they prefer the pentagons instead?
- (2) The vdw correction has not been included in the calculations. Will it influence the final results?
- (3) The corresponding "Fermi velocity" would be given for the linear dispersed bands. Are these values consistent with the experimental data?

Reviewer #2 (Remarks to the Author):

Cerda' and coworkers present an interesting study of Si nanoribbons (NRs) and nanodots on the Ag(110) surface. On the basis of DFT calculations they propose a new model which appears to be thermodynamically stable, reproduces STM images, and seems to agree with experimental data from core-level photoemission and ARPES. They term this new model "penta-silicene" and claim it is a new allotrope of silicon. While this study is certainly a step forward in our understanding of the Ag(110):Si system, I do not find it at the level of Nature Communications.

In the following I have tried to justify my decision based on the Editor's original suggested criteria.

For brevity, I define here some extra references, used below:

- R0 Colonna JPCM 25 315301 (2013)
- R1 Bernard PRB 92 045415 (2015), Satta PRL 115 026102 (2015)
- R2 Speiser APL 104 161612 (2014)
- R3 Prevot PSSB 253, 206 (2016)
- R4 Quhe Sci. Rep. 4 5476 (2014)
- R5 Setvin PRB 84 115317 (2011); An PRB 61 3006 (2000)
- R6 Stekolnikov PRL 93 136104 (2004)
- R7 Battaglia PRL 102 066102 (2009)
- R8 Dabrowski PRL 73 1660 (1994)

R9 Wippermann PRL 105 126102 (2010)
R10 Fujikawa PRL 88 176101 (2002)
R11 Erwin PRB 80 155409 (2009)
with other references numbered [1], [2] etc.

** What are the major claims:

The authors make two major claims:

1. The structure of Si nanoribbons that are long known to form on the Ag(110) surface is described by a new model based on Si atoms arranged in chains of pentagons lying inside missing rows of the Ag(110) surface.
2. This structure constitutes a new, naturally occurring, allotrope of Si: a pentagonal analog of silicene

** Are the claims novel? If not, please identify the major papers that compromise novelty

1. The precise structural model is new, but does not constitute a huge step forward, as it is built upon many previous observations and simulations:

- a. A $x2$ periodicity [13]; or $5x2/c(10x2)$ cell [R0] for double NRs;
- b. the presence of missing rows of Ag [24];
- c. Si/Ag surface reactivity/miscibility [20] on Ag(110), [R1] on Ag(111));
- d. strong Si-Ag bonds [18,24,R2,R3]
- e. short Si-Si bonds [18] or dimers [20];
- f. a somewhat similar structural model was proposed [20] that considers all of these details (discussed only in the supplementary material).
- g. free-standing silicene models [5,16] do not agree with experiments [17,20,R2,R3]

2. That the Si nanoribbons constitute a new allotrope of Si is a novel idea, but severely flawed (see below).

** Will the paper be of interest to others in the field?

The paper should be interesting to the surface science community in that it claims to have solved the question of the Si nanoribbon structure.

However, since the structure consists of a strongly interacting Si-Ag adlayer (as the strong S1 band dispersion, Si-Ag hybridization, and mixed sp^2/sp^3 bonding shows), it cannot have any of the interesting properties (Dirac cone, etc) of true silicene, as widely noted by others [R4,R3], and is therefore useless for technological applications, and of less interest to the general scientific community. (This is also reflected in the general decrease in interest in Ag110-silicene in high impact journals.) It appears to be little more than "a Ag(110) reconstruction induced by Si" [24].

** Will the paper influence thinking in the field?

If the model is further confirmed, it should put to rest the widespread (and in my opinion, weak) claims that these Si nanoribbons are composed of strips of silicene (e.g. [5,16]).

** Are the claims convincing? If not, what further evidence is needed?

1. The authors' DFT calculations are convincing (in particular the phase diagram in Fig 9), their structural models for the nano-dots and nanoribbons look very plausible, and the agreement with STM images is satisfactory.

Very little, if any, new experimental evidence is presented: old data (STM [13,15], PES [22,23], EDC [25]) are reinterpreted in light of the new model. The evidence presented is a bit weak, in my opinion, and some important data are ignored:

- The model is in strong disagreement with previous independent measurements of the Si coverages. If I calculate correctly, their model contains 12 atoms of Si per $(5x2)$ cell, much higher than the value of 8 atoms reported independently using Rutherford back scattering [24,R3] and STM/QCM [20].
- Their STM simulations of a competing model (ZZ-MR) seem in better agreement with experiment than their own model (P-MR), at least for the double NR (Fig. 8).

- While the two-component nature of the PES is unarguable (binding energies), I have less faith in the spectral intensities. Indeed, some of the authors have previously noted their strong dependence on incident energy and light polarization vector [22], and on the collection angle and photoelectron diffraction effects [23], with relative intensities changing in an unpredictable manner. The exact ratio is reported initially as "roughly 2:1" and thereafter simply "2:1" (without errors), but it seems to be much higher in their previous work [23].

2. The "penta-silicene" claims are not remotely convincing, and I believe constitute "overselling" their work:

a. The authors claim their structure is unique due to the lack of pentagonal Si rings in nature.

Actually, Si pentagonal structures have long been noted on surfaces, most notably the Si(110)-(16x2) surface [R5,R6] and Si(331)-(12x1) surface [R7,R8]. To quote [R7]: "establishing the pentamer as a universal building block for complex silicon surface reconstructions". These pentagonal chains have been observed experimentally and in simulations.

b. As in those surfaces, the pentagonal shapes are not even true pentagons of Si: at best they are perhaps "Si tetramer chains with adatoms", at worst, dimer chains and adatoms. On Si(110) the relevant pentagonal building blocks are called "adatom-interstitial-tetramers". (The authors admit as much by calling the "up" atoms Si_{ad}, and the rest Si_s). In spite of the pentagonal shaped isosurface in Fig 4a, the evidence clearly points to two kinds of Si atom, with different orbital hybridization.

c. The claim that this structure constitutes a new allotrope of Si is very exaggerated. The structure is stabilized by, and cannot exist without, the substrate: a true 2D allotrope could be the very different "silicene" [8]. Adlayers often form interesting structural motifs. For example, In adsorbed on Si(111) forms hexagon chains [R9]; Ge forms broken hexagons or horseshoes on Si [R10]; neither of these claim to be new "allotropes". Au on Si(111) forms quasi-1D chains made of dimerized gold pairs plus adatoms [R11] that have many similarities with the proposed model - but it is a "Au-stabilized surface reconstruction."

d. Last, even the term "silicene" is inappropriate. As noted above, the evidence is overwhelming that the atoms are firmly bonded to/interact with the substrate, the plane is not flat, different types of atoms are present, no Dirac cone is present, and so on. The "penta-silicene" sheet discussed in [7] is a very different beast than this Si adlayer.

** Are there other experiments that would strengthen the paper further? How much would they improve it, and how difficult are they likely to be?

1. Regarding the structural model:

- Measurements and simulation of XRD and Raman would help strengthen their model.

- Incompatibility with the reported Si coverage is an important issue to resolve.

- A deeper link between the nano-dot and nanoribbon could help explain the structural growth (e.g. hexagonal to pentagonal transition).

2. Unless the authors can synthesise stable, isolated, pentagonal ribbons, or demonstrate clear silicene-like properties, I don't see any possible justification of their second claim.

** Are the claims appropriately discussed in the context of previous literature?

Although the authors reference the history of the system by a sweeping "[13-21]" on the first page, they often do not refer to specific findings, data, or conclusions of these works appropriately. For instance, a detailed topography of the nanodot appeared in [15]; the statement "reveals the tendency of the Ag(110) surface upon Si adsorption to remove top row silver atoms" has been demonstrated in [24,R3,20]; the similar ZZ-MR model [20] is not mentioned in the main text; it is not very always clear that the STM, PES and EDC data have been previously published, as noted above; the coverage issue is ignored; "dislocation defects" (Fig. 7) were discussed extensively in [20]; and so on.

Regarding the "penta-silicene" claims, see above.

** If the manuscript is unacceptable in its present form, does the study seem sufficiently

promising that the authors should be encouraged to consider a resubmission in the future? The theoretical simulations are very good and the resulting model is quite convincing, and should certainly be published somewhere, but the overall impact of the study is not, in my opinion, adequate for Nature Communications.

Reviewer #3 (Remarks to the Author):

In this work, authors have mainly combined STM and DFT simulations to investigate the penta-silicene nature occurring on Ag(110), which gives an impressive experimental evidence about a novel low-dimensional silicon phases with pentagonal rings. I think this finding is reasonable and very important for silicon research, will definitely promote further attempts to synthesize analogous exotic low-dimensional silicon allotropes.

Some comments need to be further considered:

1. What is the possible reason for the creation of the missing row (MR), or the dislocation defects in Fig.7? In the nano-dot precursor discussion, authors give an initial judgment but not clear enough.
2. What is the main role of the distance between P-MR and Ag substrate in the investigation of its energetic stability, or its formation energy? Based on the optimized geometry in Fig.6 as well as Table I, there should be very strong interaction between silicon and Ag substrate.
3. Tables and Figs need to reorder to arrange in the context.

Reviewer #1 (Remarks to the Author):

In this paper, the authors have re-checked the geometric structure of silicene on the Ag surface. Through DFT calculations and STM comparisons, they have found that unlike the previously proposed Si hexagons, the narrow Si nanoribbons on Ag(110) surface will be pentagonal. With the help of missing Ag rows, Si atoms are localized in these troughs and form "penta-silicene" nanoribbons on the surface. In fact, such formation of pentagonal structures is similar to those in nanotubes, which is a natural result of the space constraint. The formed pentagonal Si nanoribbons also exhibit linear dispersions just below the Fermi level, which is in line with the previous experiments. The present paper gives a new interpretation for the grown Si nanostructure on Ag surface, which is helpful to understand their novel behaviors. Thus, I recommend the publication of this paper on the Nature Communications.

Some additional queries are as follows:

(1) Since the isolated Si nano-dot still keeps the hexagonal structure. How do they evolve to the pentagonal structure for extended stripes? If the initial Si10 atoms are placed in a trough (not just in an Ag di-vacancy), would they prefer the pentagons instead?

As the referee suggests, we have relaxed an isolated nano-dot hexagonal structure (NHS) after removing all Ag atoms in the missing row (MR) trough to find that although the system undergoes a strong restructuring, the hexagonal inner arrangement remains. Furthermore, placing the same number of Si atoms within a pentagonal arrangement even leads to a less stable structure (by 170 meV) because one of the pentagonal rings cannot be closed with 10 Si atoms. This was checked by attaching four Si atoms to the NHS (two at each side) and comparing its energy versus that of a cluster made of four (closed) pentagonal rings. This time the latter becomes 520 meV more stable! Further attempts to destabilize the NHS included placing two adjacent nano-dots in the same trough or even creating a strand of them but in both cases the hexagonal arrangement was preserved. However, again, the NHS structures turned out to be less stable than their pentagonal counterparts.

These results show that the precise determination of the mechanism behind the formation of the pentagonal strands from the NHS is not at all trivial and most probably involves complex exchange processes between the Si and Ag atoms. Ideally, the transition could be monitored via Variable Temperature STM experiments in order to capture (freeze) any intermediate states associated to the transition as well as by performing large scale Molecular Dynamic simulations[RR1]. Unfortunately, neither of the two approaches can be tackled within a reasonable time frame and, therefore, we cannot include a precise characterization of the transition in the present manuscript.

These new results have been added to the manuscript as Supplementary Figure 7, while we have added the following sentence in the last paragraph of the main manuscript: "At this point, however, we cannot determine the precise diffusion mechanism or even that behind the hexagonal-to-pentagonal transition (see Supplementary Figure 7)."

(2) The vdW correction has not been included in the calculations. Will it influence the final results?

We have checked that including semi-empirical van der Waals forces (see Refs. [34,35]) to our optimized P-MR models does not alter the final optimized geometries (i.e. vdW corrections to the forces were smaller than 0.01 eV/Å). This is actually

an expected result since the NRs are chemically bound to the Ag(110) surface (see Si-Ag distances in Supplementary Table II).

We have included the following sentence in the Methods section: "Test calculations showed that including van der Waals corrections [34,35] yielded negligible changes in the optimized geometries and, therefore, they have been neglected."

(3) The corresponding "Fermi velocity" would be given for the linear dispersed bands. Are these values consistent with the experimental data?

Following the referee's suggestion, we have calculated for the DNR's band structure the Fermi velocity (v_F) along Γ -Y for the linear part of the two most intense bands emerging from Γ at around -0.7 eV (see Figure RR1(a) below). We find $v_F=1.3 \times 10^6$ m/s for the band dispersing upwards (i.e. that defining the electron pocket, EP) and $v_F=1.0 \times 10^6$ m/s for the one dispersing downwards and holding a strong Si weight. The latter is in perfect agreement with the experimental value of $v_F=1.0 \times 10^6$ m/s obtained from the Si derived band shown in Figure RR1(b) (taken from Figure 12 in Ref. [now 28]). These velocities are close to that of graphene although we note that in our case the bands are hybridized with the Ag.

The Fermi velocities are now provided in the revised manuscript in the discussion on Figure 4. Additionally, in the new Supplementary Figure 8 showing the various free-standing penta-silicene band structures considered, we also provide the corresponding velocities for the bands around the Fermi level. In all cases v_F takes values of the order of 10^5 m/s for the most dispersive bands.

Figure RR1: Band structure of the Si-DNRs: (a) Figure 4(d) in the manuscript with the calculated electronic structure and, (b) experimental ARPES data extracted from Figure 12 in Ref [now 28] with the Si derived band indicated by the arrow. The circles are placed at the energies at which the v_F have been evaluated. In the rightmost panel we display the Ag(110)-(1x1) Brillouin zone (BZ) (dark rectangle) and the (5x2) BZ in an extended scheme (blue rectangles). Upon k-folding, the X-point of the (1x1)-BZ transforms to the Γ -point of the (5x2) BZ. The dashed red line indicates the direction of the measured ARPES.

Reviewer #2 (Remarks to the Author):

Cerda' and coworkers present an interesting study of Si nanoribbons (NRs) and nanodots on the Ag(110) surface. On the basis of DFT calculations they propose a new model which appears to be thermodynamically stable, reproduces STM images, and seems to agree with experimental data from core-level photoemission and ARPES. They term this new model "penta-silicene" and claim it is a new allotrope of silicon. While this study is certainly a step forward in our understanding of the Ag(110):Si system, I do not find it at the level of Nature Communications.

In the following I have tried to justify my decision based on the Editor's original suggested criteria.

For brevity, I define here some extra references, used below:

R0 Colonna JPCM 25 315301 (2013)
R1 Bernard PRB 92 045415 (2015), Satta PRL 115 026102 (2015)
R2 Speiser APL 104 161612 (2014)
R3 Prevot PSSB 253, 206 (2016)
R4 Quhe Sci. Rep. 4 5476 (2014)
R5 Setvin PRB 84 115317 (2011); An PRB 61 3006 (2000)
R6 Stekolnikov PRL 93 136104 (2004)
R7 Battaglia PRL 102 066102 (2009)
R8 Dabrowski PRL 73 1660 (1994)
R9 Wippermann PRL 105 126102 (2010)
R10 Fujikawa PRL 88 176101 (2002)
R11 Erwin PRB 80 155409 (2009)
with other references numbered [1], [2] etc.

**** What are the major claims:**

The authors make two major claims:

1. The structure of Si nanoribbons that are long known to form on the Ag(110) surface is described by a new model based on Si atoms arranged in chains of pentagons lying inside missing rows of the Ag(110) surface.
2. This structure constitutes a new, naturally occurring, allotrope of Si: a pentagonal analog of silicene

We are glad that the referee finds "interesting" our work and that it represents a "step forward in our understanding of the Ag(110):Si system". However, he/she raises many concerns in basically all "Editor's original suggested criteria" items, to conclude that the manuscript is not "at the level of Nature Communications". Below we attempt to answer all the concerns in the most rigorous way we can, although we are aware that some of them are very subjective and, thus, difficult to refute.

**** Are the claims novel? If not, please identify the major papers that compromise novelty**

1. The precise structural model is new, but does not constitute a huge step forward, as it is built upon many previous observations and simulations:

- a. A $x2$ periodicity [13]; or $5x2/c(10x2)$ cell [R0] for double NRs;
- b. the presence of missing rows of Ag [24];
- c. Si/Ag surface reactivity/miscibility [20] on Ag(110), [R1] on Ag(111));
- d. strong Si-Ag bonds [18,24,R2,R3]
- e. short Si-Si bonds [18] or dimers [20];
- f. a somewhat similar structural model was proposed [20] that considers all of these details (discussed only in the supplementary material).
- g. free-standing silicene models [5,16] do not agree with experiments [17,20,R2,R3]

The pentagonal missing row (P-MR) structure here determined is, to our knowledge, the first Si-based system made **solely** of Si pentagons. We do believe this claim is novel, although whether it is a huge, big or significant step forward is obviously subjective. Items (a-e) comprise previous reported facts on this system (or related ones) but we cannot understand why they are used here to reduce the novelty of our work; despite of all this previous partial information, the actual structure of the nano-ribbons on Ag(110) has remained elusive for over a decade. SiNRs on Ag(110) involved over 463 papers and around 22428 citations [RR2]. Our main achievement is

not only to have solved it but, more importantly, in doing so, discover an unprecedented pentagonal arrangement of the Si atoms. Regarding item (f), the zig-zag model presented in Ref [20][now 22] just shares with the pentagonal one the presence of Si atoms in the MR troughs, but overall it is clearly different as the number of Si atoms is different and it holds no pentagons (a simple visual comparison between Figures 3(a) and Supplementary figure 2(c) in the manuscript makes such difference clear enough). Even more, the ZZ-MR model is flawed as demonstrated by the XPS analysis and total energy calculations and we therefore do not find any strong reason for discussing it in the main text, as it would only confuse the reader. Finally, we do not understand how issue (g) can be interpreted as a criticism to our work; *silicene* on its own is only mentioned in the introduction in order to recall the large attention that novel Si-based structures are currently raising. Otherwise, we employ the term *penta-silicene* in analogy to *penta-graphene* but, as is well-known, *graphene* and *penta-graphene* are markedly different [RR3,1]. Furthermore, we are unable to follow the referee's remark in as much as our *penta-silicene* structures are one-dimensional (1D) as opposed to the 2D nature of *silicene*.

2. That the Si nanoribbons constitute a new allotrope of Si is a novel idea, but severely flawed (see below).

In order to convey with the structure of the referee's remarks, we find it more convenient to comment on this issue below.

*** Will the paper be of interest to others in the field?*

The paper should be interesting to the surface science community in that it claims to have solved the question of the Si nanoribbon structure.

However, since the structure consists of a strongly interacting Si-Ag adlayer (as the strong S1 band dispersion, Si-Ag hybridization, and mixed sp²/sp³ bonding shows), it cannot have any of the interesting properties (Dirac cone, etc) of true silicene, as widely noted by others [R4,R3], and is therefore useless for technological applications, and of less interest to the general scientific community. (This is also reflected in the general decrease in interest in Ag(110)-silicene in high impact journals.) It appears to be little more than "a Ag(110) reconstruction induced by Si" [24].

The referee is right stating that the interaction between the Si pentagons and the Ag substrate is strong (this is already obvious from Supplementary Tables I and II where Si-Ag bonding distances as well as Si adsorption energies are provided, or from the Si-Ag hybridized bands also discussed in the text, as the referee correctly recalls). Nevertheless, at this point we cannot anticipate if the electronic properties of the pentagonal NRs are "useless for technological applications", and we are particularly surprised that the referee is so confident in this respect because "it cannot have any of the interesting properties of true *silicene*". We make no claim (or even suggest) that the NRs share similar properties with those of *silicene* (even their dimensionalities are different!). Regarding the pessimistic point of view adopted by the referee, we recall a few thoughts:

i) Would the electronic structure of the pentagonal NRs on Ag(110) had already shown exciting properties (gapless Dirac cones, a non-trivial topology or reduced dimensionality many-body related phenomena, for instance) we would be facing a rather extraordinary discovery which, in our opinion, should merit publication in the highest ranked scientific journals (with impact factors > 30).

ii) The work presented here should be considered as an initial milestone in the synthesis of quasi-perfect Si-based 1D pentagonal structures -indeed, that is why we consider the *penta-silicene* structure as a breakthrough. In this sense, we find it perfectly conceivable that future work along this research line may lead to analogous structures with very promising properties which may additionally exploit their 1D dimensionality.

iii) Carbon-based pentagons and the coexistence of sp²/sp³ bonds are attracting great research activity since they favor larger spin-orbit (SO) effects [RR4]. Given the larger atomic weight of silicon versus carbon, we would therefore expect

that any SO coupling should be enhanced in our NRs, thus opening the possibility to access a wealth of exotic phenomena.

iv) In Nanotechnology words such as "useless" are questionable. For instance, a transistor based on silicene grown on Ag(111) ---i.e. completely detached except for two pads to make the contacts-- has been recently realized [RR5]. In this line, the Si NRs should be highly compatible with current Si-based technological devices.

*** Will the paper influence thinking in the field?*

If the model is further confirmed, it should put to rest the widespread (and in my opinion, weak) claims that these Si nanoribbons are composed of strips of silicene (e.g. [5,16]).

We fully agree with the referee that our model "puts to rest" all of the previous ones (including [5] and [16][now 7 and 18]). But, as stated above, the figure of merit of our paper in this sense is that it represents the first realization of a Si-based structure made up solely of pentagons, and this fact will probably promote multiple experimental and theoretical studies aiming to achieve and characterize similar/alternative pentagonal structures beyond carbon.

*** Are the claims convincing? If not, what further evidence is needed?*

1. The authors' DFT calculations are convincing (in particular the phase diagram in Fig 9), their structural models for the nano-dots and nanoribbons look very plausible, and the agreement with STM images is satisfactory.

Very little, if any, new experimental evidence is presented: old data (STM [13,15], PES [22,23], EDC [25]) are reinterpreted in light of the new model. The evidence presented is a bit weak, in my opinion, and some important data are ignored:

- The model is in strong disagreement with previous independent measurements of the Si coverages. If I calculate correctly, their model contains 12 atoms of Si per (5x2) cell, much higher than the value of 8 atoms reported independently using Rutherford back scattering [24,R3] and STM/QCM [20].

We were well aware of the discrepancy on the Si coverage, but preferred to exclude a discussion on it in the manuscript since we are confident that the measurements of the coverage hold much larger errors than those reported in the works cited by the referee. It is very well known in the Surface Science community that submonolayer coverage estimates using quartz crystal microbalance (QCM) measurements are prone to very large errors and uncertainties. The referee cites a work where, based only on QCM, a rough estimate of 8 +/-2 Si atoms per (5x2) cell was obtained (ref. [20](now [22])). To get such relatively small error (25% error), the authors performed a careful job, trying to minimize the sources of error, maintaining a constant temperature of the QCM by refrigerating it with water, and averaging measurements for very long times. However, some further serious error sources have clearly not been taken into account. For instance, according to the evaporation rate in Fig. 2 of ref. [20] (now [22]), the authors performed the Si deposition on Ag(110) for much shorter times (of the order of 15 s) than the times used for averaging (of the order of 1500 s). This huge difference could account for very different rates with much larger deviations for the quoted average value. Moreover, the measurements were performed with a quartz kept at 7.7°C but not on the Ag(110) surface used in the experiments that was heated above RT at 202°C while depositing Si. This means that the sticking coefficients of the cold quartz and the hot Ag(110) could be very different. All these factors, and others, could give systematic errors much larger than those quoted. So it is not strange that a nominal coverage of 8 atoms per (5x2) (instead of 12 according to our model), could have been mistakenly measured by QCM. A systematic error of 50% in the measurements would yield a perfect agreement with our coverage.

On the other hand, RBS measurements can yield very valuable information regarding the composition of a sample. However, in the reference cited by the referee [24](now [26]) and R3], RBS was not used in situ under UHV on the Si/Ag(110) that was studied. RBS was used ex situ as a means of estimating the calibration of the Si evaporator. Thus, these measurements can only be considered as an estimation of

the evaporator flux, as it is well known that evaporator rates can have fluctuations in space and time that can yield large variations in coverage determinations performed *ex situ*.

Last, we recall that Surface Science history has plenty of examples where "accurate" determination of coverages have resulted in being not so accurate. Just to cite an example for illustrative purposes, the Pb/Si(111) system presents several reconstructions close to 1ML whose exact nature and coverage have been the subject of intense debate for decades. In particular, the so-called Pb/Si(111)-R7xR3 phase was conjectured to have a coverage of 1 ML by Hwang et al. [RR6] based on previous RBS measurements[RR7]. However, all first-principles calculations and most other experimental evidences have shown that the correct coverage is 1.2 ML [RR8]. And still, the debate regarding the coverage of this phase seems to be open.

- Their STM simulations of a competing model (ZZ-MR) seem in better agreement with experiment than their own model (P-MR), at least for the double NR (Fig. 8).

From our experience over decades on STM experiments and simulations as well as their quantitative comparison, we regard that the "better agreement" invoked by the referee is completely fortuitous. Let us first recall that STM simulations represent a very useful technique in structural characterizations but, as already stated in the text, it is well established that different models may yield almost identical images [RR9] and, in such cases, the simulations only serve to corroborate a proposed model based on other facts. This is precisely the case here, where the ZZ-MR and P-MR models lead to similar simulated images (both correlating well with experiment), but total energy calculations and XPS analysis unambiguously point to the P-MR structure as the correct one, so that the STM simulations just confirm the goodness of the model.

In Figure RR2(a) below we reproduce a zoom-in of our own experimental DNR STM image (i.e. Figure 1(c) in the manuscript), in (c) a very similar one acquired at 77K taken from Ref [21](now [23]) and in (b) the simulated one for the P-MR model (Figure 3(g) in the manuscript). In our opinion, any expert in the field would regard the experiment-theory agreement as excellent and convincing (i.e. it fully corroborates the P-MR model deduced from the DFT and XPS analysis). The simulations for the two possible DNR phases under the ZZ-MR model (Figure 8(c)) indeed also match very nicely the experimental figures (a) and (c) but both structures can be safely ruled out due to their low stability and their 1:1 ratio between the Si_s and Si_{ad}. In fact, the differences among all images are well below the uncertainties associated to the actual simulation, which are mainly two: (i) the exact tip termination (the great unknown in STM experiments) and, (ii) the many approximations assumed in the theoretical approach (despite we regard our employed formalism as state-of-the-art). Therefore, the "better agreement" of the ZZ-MR model cannot be used as an argument to question the validity of the P-MR model.

Figure RR2: STM images of the Si DNRs. (a) and (b) Experimental (at RT) and theoretical images presented in this work, respectively and, (c) experimental image (at 77 K) reported in Ref.[21] (now [23]).

Apart from the subtle differences in the aspect of the images, the referee may have also noticed that the calculated corrugation (1 Å) is larger than the experimental one (0.4 Å). If so, we are actually glad that he/she brings in this issue, since this is the first contribution on Si NRs where experimental STM line profiles are quantitatively compared against theory. As shown in Figure RR3 below, if the simulation is carried out at the experimental bias of 1.3 Volt, a smaller overall corrugation is obtained (0.7 Å), only 0.3 Å larger than the experimental one. Moreover, we do not believe that a general reader of Nature Communications will be interested in such fine details and we prefer not to introduce a discussion on the dependence of the corrugations on the bias in the manuscript.

Figure RR3: STM simulated topographic image for the DNR P-MR structure calculated for set points $V=1.3$ V and $I=1$ nA. The associated profile along the bumps is displayed at the right (solid blue line) together with the same profile obtained for $V=-0.2$ V (dashed line) --i.e. the same as in Figure 3(g).

- While the two-component nature of the PES is unarguable (binding energies), I have less faith in the spectral intensities. Indeed, some of the authors have previously noted their strong dependence on incident energy and light polarization vector [22], and on the collection angle and photoelectron diffraction effects [23], with relative intensities changing in an unpredictable manner. The exact ratio is reported initially as "roughly 2:1" and thereafter simply "2:1" (without errors), but it seems to be much higher in their previous work [23].

The referee is correct stating that in PES the peaks are affected by matrix elements while polarization and/or photoelectron diffraction effects can also

influence the spectra associated to particular electronic states. However, PES is a reliable technique for relating the intensity of a peak with the number of electrons of equivalent energy ejected during the experiment [RR10]. In our case, we are confident about the 2:1 ratio after checking that a similar proportion is preserved for different photon energies under the same geometry with only small deviations inherent to the photoemission process. We provide in Figure RR4(a-b) below two spectra acquired at very different photon energies. The relative area of the two components is 2.39:1 at 135.8 eV and exactly 2.0:1 at 700 eV. Since diffraction effects should be markedly different between the two spectra: at low photon energy (135.8 eV) backscattering effects will dominate while at high photon energies the forward scattering processes become dominant, this represents a clear indication that the relative occupancy of the two different Si components should be close to 2:1 with a maximum error of just 0.4. Furthermore, we have digitalized and next analyzed the spectra reported in Ref. [23] (now [25]), also acquired at normal emission but in a different synchrotron and beamline (Elettra synchrotron and VUV beamline). The deconvolution, presented in Figure RR4(c), shows basically the same ratio (2.2:1) as that deduced here, demonstrating the robustness of the PES data. Given that the referee requires an explicit error, we have added the following statement in the discussion on Fig. 1d: "(we estimate a maximum error of 20% in the $Si_s:Si_{ad}$ intensity ratio based on analogous spectra recorded at different energies or even beamlines [25])."

We finally note that very similar ratios in the PES intensity analysis have been obtained for the extensively studied $\sqrt{3}\times\sqrt{3}$ -Sn/Ge(111) system (Ref. [RR11]), where two Sn atoms at two different positions yield roughly the same 2:1 ratio regardless of the photon energy or the geometry.

Figure RR4: Normal emission Si 2p core level spectra for the DNRs on Ag(110) recorded at (a) 135.8eV (same as Fig 1d in the manuscript), (b) 700eV photon energy and (c) 135.8eV (digitalized from [23] (now [25])).

2. The "penta-silicene" claims are not remotely convincing, and I believe constitute "overselling" their work:

a. The authors claim their structure is unique due to the lack of pentagonal Si rings in nature. Actually, Si pentagonal structures have long been noted on surfaces, most notably the Si(110)-(16x2) surface [R5,R6] and Si(331)-(12x1) surface [R7,R8]. To quote [R7]: "establishing the pentamer as a universal building block for complex silicon surface reconstructions". These pentagonal chains have been observed experimentally and in simulations.

We thank the referee for pointing to us these two complex Si reconstructions which we were not aware of. Accordingly, we have modified the introductory paragraph in order to cite them. However, after a visual inspection of our 1D P-MR model and the highly complex 2D reconstructions deduced in [R5-R8], it is clear that they hold no resemblance -see Fig. RR5 below. The Si(110)-(16x2) model comprises a large amount

of different Si polygons, including only a few quite irregular pentagons. On other hand, the pentamer motif common to both structures contains 6 Si atoms and is markedly different than the pentagons in our P-MR structure! Therefore, we cannot understand the referee's statement: "These pentagonal chains have been observed experimentally and in simulations".

Figure RR5: Top views of (a) the Si(110)-(16x2) surface reconstruction extracted from Stekolnikov et al, PRL 93 136104 (2004), (b) the Si(331)-(12x1) surface reconstruction extracted from Battaglia et al, PRL 102 066102 (2009) and, (c) the P-MR model deduced in the present work.

b. As in those surfaces, the pentagonal shapes are not even true pentagons of Si: at best they are perhaps "Si tetramer chains with adatoms", at worst, dimer chains and adatoms. On Si(110) the relevant pentagonal building blocks are called "adatom-interstitial-tetramers". (The authors admit as much by calling the "up" atoms Si_{ad} , and the rest Si_s). In spite of the pentagonal shaped isosurface in Fig 4a, the evidence clearly points to two kinds of Si atom, with different orbital hybridization.

Again, the referee is completely correct stating that they are not "perfect" pentagons; this is indeed extensively discussed in the manuscript and Supplementary figure 3, where all the structural details are given. This is a typical situation when placing a defined cyclic structure on a surface in which case, and unless the interaction is very weak, atoms with different environments present different chemical shifts. Among many others we may point out a few examples: (i) for C_{60} on Pt(111) one also finds a chemical shift between those carbons linked or not to the surface [RR12], (ii) silicene on Ag(111), due to the puckered conformation of the Si hexagons on the surface, also shows a chemical shift between the silicon atoms [RR13], or (iii) graphene chemisorbed, either in registry with the surface or forming a moiré pattern [RR14], can become strongly buckled with carbon atoms presenting different reactivities depending on their registry with respect to the substrate. However, the allotropic form either of C_{60} or the Si and C hexagons is preserved in each case. Once we have proved that the free-standing 1D *penta-silicene* phase is metastable (see new Supplementary figure 8 in the revised manuscript) we believe, without any loss of rigour, that the same concepts can be applied here since we essentially find a 1D *penta-silicene* allotrope adsorbed on the Ag(110) surface.

As a second counterargument against the referee's remark, we may disregard the substrate and focus on a cyclopentane molecule, which is recognized in chemistry as a C-based pentagonal structure. It is well known that its five-membered ring adopts a non-planar puckered conformation, denoted as "envelope", in order to reduce the eclipsing and torsional strains. As shown in the comparative Figure RR6 below, this envelope conformation is indeed very reminiscent of that of the pentagons in the P-MR structure or even in one of its (hypothetical) hydrogenated free-standing phase.

Figure RR6: Perspective views of (a) a cyclopentane molecule in the flat and the more stable envelope conformations, (b) the P-MR model with the Ag atoms drawn as small balls and, (c) a free-standing penta-silicene strand with the edge Si_{ad} saturated with a H atom (see Supplementary Figure 8 for further details).

c. The claim that this structure constitutes a new allotrope of Si is very exaggerated. The structure is stabilized by, and cannot exist without, the substrate: a true 2D allotrope could be the very different "silicene" [8]. Adlayers often form interesting structural motifs. For example, In adsorbed on Si(111) forms hexagon chains [R9]; Ge forms broken hexagons or horseshoes on Si [R10]; neither of these claim to be new "allotropes". Au on Si(111) forms quasi-1D chains made of dimerized gold pairs plus adatoms [R11] that have many similarities with the proposed model - but it is a "Au-stabilized surface reconstruction."

It is obviously true that the Si pentagons in the P-MR model are stabilized by the Ag(110) and, in this sense, we admit the term "allotrope" could be arguable (see also reply to point b. above). Theoretically, we have optimized free-standing penta-silicene NRs together with alternative hydrogenated configurations to find that all of them are indeed metastable and preserve the pentagonal rings. The LDA value for the cohesive energy of pristine penta-silicene is 4.9 eV/atom which amounts to almost 80% of the computed adsorption energy for the P-MR structure (6.4 eV/atom --see Supplementary Table I) implying that, energetically, the main stabilization mechanism is the Si-Si pentagonal bonding rather than the interaction with the Ag(110) substrate. The results have now been included in the manuscript as new Supplementary Figure 8. We have even found that saturating the Si_{ad} atoms by just one H atom leads to a restructuring of the pentagons highly reminiscent of that found for the P-MR model (compare Figs. RR6(b) and RR6(c)). These new results put hope on the possibility to realize quasi-free-standing penta-silicene in the near future. In any case, since the claim that we have discovered a novel 1D Si allotrope appears only once in the manuscript and in order to avoid controversies/disputes about "overselling" our work, we have rewritten the last sentence of the conclusion, which now reads: "The discovery of this novel Si-based pentagonal phase puts hope on the realization of a new 1D Si allotrope via hydrogenation and/or weakening its interaction with the substrate in order to obtain quasi-free standing penta-silicene (see Supplementary Figure 8 for a summary of the atomic and electronic structure of free-standing penta-silicene). We are also convinced that our study will promote the synthesis of analogous exotic Si phases on alternative templates with promising properties[10])."

d. Last, even the term "silicene" is inappropriate. As noted above, the evidence is overwhelming that the atoms are firmly bonded to/interact with the substrate, the plane is not flat, different types of atoms are present, no Dirac cone is present, and so on. The "penta-silicene" sheet discussed in [7] is a very different beast than this Si adlayer.

It seems that the referee keeps on assuming that by "penta-silicene" we mean

"silicene", but there is no scientific reason for doing so. We again recall that "penta-graphene" is markedly different than "graphene", and so is "silicene" with respect to what we have named here "penta-silicene". Should the referee propose a more appealing/appropriate name for this novel 1D phase, we would be glad to incorporate and acknowledge it.

*** Are there other experiments that would strengthen the paper further? How much would they improve it, and how difficult are they likely to be?*

1. Regarding the structural model:

- Measurements and simulation of XRD and Raman would help strengthen their model.

We are not acquainted either with XRD simulations or with the Raman technique. Therefore, performing such kind of experiments (or establishing appropriate collaborations) would delay the publication of our work for over a year, something we find unnecessary given the strong evidences that support the P-MR model. We are aware, however, that an XRD group will test our P-MR model based on their previously acquired data [RR15].

- Incompatibility with the reported Si coverage is an important issue to resolve.

We believe we have clearly explained above that there is no such "incompatibility", but simply a large underestimation of the error bars in the reported Si coverages.

- A deeper link between the nano-dot and nanoribbon could help explain the structural growth (e.g. hexagonal to pentagonal transition).

This is a very interesting issue to explore but would also require time scales beyond one year, while the actual mechanism of the transition is clearly beyond the scope of the current work. We nevertheless prove, after the results shown in the new Supplementary Figure 7 that pentagonal moieties with the edge pentagons closed are clearly more stable than their hexagonal counterparts.

2. Unless the authors can synthesise stable, isolated, pentagonal ribbons, or demonstrate clear silicene-like properties, I don't see any possible justification of their second claim.

Synthesizing "stable, isolated, pentagonal ribbons" such as those presented in the new Supplementary Figure 8 would be an extraordinary breakthrough in Si chemistry (again, not necessarily presenting silicene-like properties). Still, and as stated above, we hope our current work will motivate multiple works aiming to realize quasi-free standing Si pentagonal structures. Obvious research routes to achieve it could be hydrogenation and/or replacing the Ag(110) surface by a different (less reactive) template. We are also aware of STM experiments where the NRs could be partly lifted off the surface by the tip [RR16].

*** Are the claims appropriately discussed in the context of previous literature?*

Although the authors reference the history of the system by a sweeping "[13-21]" on the first page, they often do not refer to specific findings, data, or conclusions of these works appropriately. For instance, a detailed topography of the nanodot appeared in [15]; the statement "reveals the tendency of the Ag(110) surface upon Si adsorption to remove top row silver atoms" has been demonstrated in [24,R3,20]; the similar ZZ-MR model [20] is not mentioned in the main text; it is not very clear that the STM, PES and EDC data have been previously published, as noted above; the coverage issue is ignored; "dislocation defects" (Fig. 7) were discussed extensively in [20]; and so on.

Due to the limited space, we find it impossible to go in detail over the Si NRs' history or describe all the specific data reported so far. We leave such discussion to a review paper which is in progress, while we find much more interesting to the Nature Communications reader to focus on the characterization of this novel structure. In fact, we do not understand why the referee insists on discussing in the main text about the model proposed in [20](now[22]) which is clearly flawed. On the other hand, we have changed a few captions to make sure there is no doubt about which data has been reported before or was unpublished. All presented STM images

are unpublished data while the Si 2p core level in Fig. 1d has been adapted from Fig.1e in [22] (now [24]) although in that work it was not deconvoluted. The two EDC curves in Fig. 4b have been adapted from Ref. [22] and [25] (now [24] and [28]). An explicit mention to Ref.[20] (now [22]) has been included in the discussion on Supplementary Figure 6 while we believe that a proper citation to Ref. [15] (now [17]) was already included in the main text: "The images are in perfect accord with previous works [15, 17, 22, 23]." Regarding the coverage issue, we feel it is more elegant and less misleading to ignore it instead of stating the large errors associated to the measurements in [24 (now [26]), R3,20 (now [22])], as discussed in detail above. Finally, we agree with the referee that the discussion on the removal of top Ag atoms by Si should be better contextualized and, accordingly, we have rewritten the sentence: "Furthermore, it reveals the tendency of the Ag(110) surface upon Si adsorption to remove top row silver atoms, i.e. the initial stage in the creation of a missing row (MR) and incorporate Si nanostructures in the troughs." to "... and confirms the tendency of the Ag(110) surface to remove top row silver atoms upon Si adsorption [22,26], as could be expected from the low stability of this particular surface [27]."

Regarding the "penta-silicene" claims, see above.

*** If the manuscript is unacceptable in its present form, does the study seem sufficiently promising that the authors should be encouraged to consider a resubmission in the future?*

The theoretical simulations are very good and the resulting model is quite convincing, and should certainly be published somewhere, but the overall impact of the study is not, in my opinion, adequate for Nature Communications.

We hope that the arguments given above to the Referee's remarks will make him/her reconsider the overall impact and reliability of our study.

Reviewer #3 (Remarks to the Author):

In this work, authors have mainly combined STM and DFT simulations to investigate the penta-silicene nature occurring on Ag(110), which gives an impressive experimental evidence about a novel low-dimensional silicon phases with pentagonal rings. I think this finding is reasonable and very important for silicon research, will definitely promote further attempts to synthesize analogous exotic low-dimensional silicon allotropes.

Some comments need to be further considered:

1. What is the possible reason for the creation of the missing row (MR), or the dislocation defects in Fig.7? In the nano-dot precursor discussion, authors give an initial judgment but not clear enough.

The reason is the well-reported low stability of the pristine Ag(110) surface [27]. In this line, we have performed additional calculations to address the energetic balance of exchanging a top silver atom with a nearby Si atom in order to develop the pentagonal missing row strand. As shown in Figure RR6(a) below, we consider a hypothetical 4-pentagon structure resting within a 5 Ag vacancy, while an additional Si atom, Si_e , is placed in an adjacent hollow site (marked by a yellow dot). Upon interchanging the silver top row atom just below the Si cluster (Ag_t , marked with a black dot) by the Si_e we find two metastable configurations: (b) a less stable one (by 68 meV) where the Si_e ends up in a short bridge site and makes a bond with the Ag_t and (c) a second one where the Si_e occupies the hollow site left by the Ag_t and which is as stable as the initial configuration. Therefore, the calculation reveals that there is no energy penalty in the energetic balance for the exchange process, while configuration (b) could be regarded as an intermediate step. Obviously, this proposed path is purely hypothetical, and one may expect that

if instead of one, three Si_e atoms were exchanged with just one Ag_t then the energy balance would clearly favor the new 5-pentagon structure -recall that the new results shown in Supplementary Figure 7 indicate that close pentagon clusters are much more stable than those with open pentagons.

Figure RR6: A hypothetical path for the exchange process between a silver atom (Ag_t) and a nearby Si adatom (Si_e) in order to develop the pentagonal MR structure. Numbers on top of the arrows give the total energy differences (positive less stable) with respect to (a), which is found as the most stable one.

Still, since we cannot answer the precise exchange mechanism, we have added the following statement in the main manuscript: "At this point, however, we cannot determine the precise diffusion mechanism or even that behind the hexagonal-to-pentagonal transition (see Supplementary Figure 7)." Likewise, at this point we cannot precise the origin of the dislocations, but can only provide a few hints based on the STM images. Accordingly we have added the following text to the discussion on Supplementary Figure 6 "Indeed, this type of dislocations are typically found when a SNR appears between two DNRs, in which case the (5x2) periodicity is truncated. Since from the STM images there does not seem to exist a point defect at the dislocation which could drive the generation of the shifted MRs, we speculate that each of them is created far away and upon further growth the MRs meet thus forming the dislocation."

2. What is the main role of the distance between P-MR and Ag substrate in the investigation of its energetic stability, or its formation energy? Based on the optimized geometry in Fig.6 as well as Table I, there should be very strong interaction between silicon and Ag substrate.

Indeed there is a strong interaction between the P-MR and the Ag substrate and, therefore, we may regard the distance between them as a source of stabilization of the *penta-silicene*, as the $\text{Ag}(110)$ surface acts as a template. Nevertheless, once the P-MR structure has been demonstrated, as well as the fact that free-standing pentagonal NRs (hydrogenated or not) are found to be metastable, at least at the DFT level (see new Supplementary Figure 8), several possible ways to detach or decrease the NR's interaction with the substrate can be conceived; for instance via hydrogenation and/or replacing the $\text{Ag}(110)$ surface with a different (less reactive) template. We also note that our LDA value for the *penta-silicene*'s cohesive energy is 4.9 eV/atom which amounts to almost 80% of the computed adsorption energy for the P-MR structure (6.4 eV/atom --see Supplementary Table I) implying that, energetically, the main stabilization mechanism is the Si-Si pentagonal bonding rather than the interaction with the $\text{Ag}(110)$ substrate.

3. Tables and Figs need to reorder to arrange in the context.

We apologize for this fact and have converted the manuscript into word format and rearranged some figures with the hope that its layout has been improved.

References (not included in the manuscript):

- [RR1] See for instance: R. Bernard et al, PRB 92 045415 (2015); M. Satta et al, PRL 115 026102 (2015); S. Maier et al, Phys. Rev. Lett. 112, 126101 (2014).
- [RR2] BY SCOPUS WEB: "ag 110" AND (silicon OR silicene) AND (nanoribbons OR nano-ribbons OR nanowires)].
- [RR3] A. H. Castro Neto et al, Rev. Mod. Phys. 81, 109 (2009).
- [RR4] See for instance: Shiheng Liang et al, Sci. Rep. 6 19461 (2016).
- [RR5] Tao L. et al. Nature Nanotechnology 10, 227-231 (2015)
- [RR6] Hwang et al, PRL 93,106101 (2004); Hwang et al, PRB 83, 134119 (2011).
- [RR7] Ganz et al, Surf. Sci. 257,259 (1991).
- [RR8] Kumpf et al, Surf Sci 448, L213 (2000); Brochard et al, PRB 66,205403 (2002); Hupalo et al, PRL 90, 216106 (2003); Kim and Yeom, PRL 107,136402 (2011); Man et al, PRL 110,036104 (2013); etc.)
- [RR9] See for instance: Carlisle C. I. et al., PRL 84, 3899 (2000); Schnadt J. et al, PRL 96, 146101 (2006).
- [RR10] Seah M. P., Surface And Interface Analysis, Vol. 2, No. 6, (1980), S. Evans, Surface And Interface Analysis, Vol. 18, 323-332 (1992).
- [RR11] Avila J. et al. J. Surface Science 433-435 327-331 (1999), Petaccia L. et al. PRB, Volume 63, 115406 (2001).
- [RR12] Martín-Gago, JA Nat. Chem. 3, 11-12 (2011), Otero G. et al. Nature Letter 454, 865-868 (2008) | doi:10.1038/nature07193).
- [RR13] Vogt P. PRL 108, 155501 (2012), Lin C-H, Applied Physics Express 5, 045802 (2012).
- [RR14] Odahara G. et al. Surface Science, 605, 1095-1098 (2011), Marchini S. et al., PRB 76, 075429 (2007).
- [RR15] Prof.Laurence Masson du CINaM, Prof. Aix-Marseille Univ. New XRD analysis (private communication).
- [RR16] Prof. Noriaki Tagaki (Tokyo University), private communication.

Reviewers' comments:

Reviewer #1 (Remarks to the Author):

After carefully checking the authors' replies, I think they have tried their best to revise the manuscript. Since the present paper has proved a new geometrical structure for low-dimensional Si nanostructures, it is helpful for the developing of silicene research. This work will be interesting for readers. Thus, I recommend this paper to be published.

Reviewer #2 (Remarks to the Author):

Firstly - my apologies to the authors and the editor for being a little late with my report due to a family illness.

In the resubmitted version of the manuscript, the authors have addressed quite well many of the reviewers' issues. (For clarity, I am the original Reviewer #2.)

In response to Reviewer #1, they have attempted to explain the hexagonal-pentagonal transition from seed to nanoribbon (NR), unfortunately without success, and report detailed results in the Supplementary Material. This is interesting work and a nice addition. They have also tested the importance of van der Waals corrections, finding them negligible (incidentally, this finding was also reported for other NR models in Ref. 22), in support of their total energy calculations. Last, they report fermi velocities for some of their computed bands, which compare well with the experimental values. While this is further support of their model, one must be careful not to infer too much about the electronic nature of these bands, which can arise from adsorbate-induced umklapp processes, as shown in Gori et al, J. Appl. Phys. 114, 113710 (2013).

In response to Reviewer #3, the authors present an example of a Si/Ag exchange process on the (110) surface, showing that such a process is energetically viable (a finding reported in Ref. 22, now referred to correctly), and comment on the observed NR dislocations. They also discuss the new Supp. Fig. 8, i.e. stability of isolated NRs versus the adsorbed system, and report that the average adsorption energy is dominated by the internal cohesive energy. Although this is a simplification (see below), it is an indication that the NRs might be detached from the surface.

At this point I feel that the authors have thoroughly addressed the issues raised by Reviewers 1 and 3.

Considering now the response to Reviewer #2 comments (mine). My initial report was very long and detailed, and I appreciate that the authors have considered each point, sometimes agreeing and sometimes rebutting them. They offer some new calculations in the supplementary material but have not made any major changes or additions to the body of the paper. Below I will try to condense the review into two points, which should make my judgement also more clear to the editors.

First (the easier point): have the authors adequately shown that their pentagonal (P-MR) structural model is correct?

In my opinion: yes. In brief, I accept their arguments that the (independent) coverage measurements may be systematically wrong; that the STM images are well explained by the P-MR structure; that the 2:1 ratio from PES is a valid measurement now that the errors have been explicitly reported; and that the evidence presented is adequate for publication. I accept that more experimental analysis is beyond the authors current scope and timeframe. The new calculations studying the hexagon-pentagon transition are a welcome addition, even though the matter remains unexplained. References to previous works have been improved, making the novel aspect of the present work more clear to the reader. Overall, therefore I am quite satisfied with the main content of the paper from page 1 (line 34) to page 3 (line 147): this text, which constitutes the

bulk of the paper, puts forward a strong case for solving the structure of the Si NRs on Ag(110). The methodology is sound and the supplementary material is supportive, although it does not push the paper to a higher level of impact.

Second: are their findings worthy of publication in Nature Communications?

In my opinion: no. I believe that the authors have identified and solved, to quote again from Ref 26: "a Ag(110) reconstruction induced by Si", but fail to demonstrate its general importance. In their rebuttal they claim that many of my objections are "subjective", but frankly, the onus is on them to show that their model--certainly aesthetically pleasing and, in a limited sense, unique--is worthy of publication in such a wide interest, high impact journal. The justification put forth is mostly contained in the abstract (page 1) and the last six lines of the main text.

Their most important claims in the manuscript are

- the work "puts hope on the realization of a new 1D Si allotrope" and (in the rebuttal) that "we hope our current work will motivate works aiming to realise quasi-free standing Si pentagonal structures"

If the authors succeeded in realizing a new Si allotrope, then, as they say in their rebuttal, they would be looking at a major scientific discovery. Without this, I fail to see anything remarkable about the system, except that the silicon therein is "solely comprising pentagonal rings". As the authors were not impressed by my citing of Si pentamer (isolated pentagons) formation on vicinal Si surfaces, I recall again the well studied case of In chains on Si(111). In that system, In forms chains of trimers or hexagons, and the system undergoes interesting phase transitions between the two. Here "phase" refers to the Si-In ensemble, not to the In chain structure. But I have not heard of any claims of "novel hexagonal In allotropes". It is simply not correct to consider only the silicon part of the reconstruction and claim it is a "silicon phase". There are many such adsorbate-stabilized 1D and 2D surface reconstructions: some reveal remarkable physical properties, and some do not. What is the remarkable physical property here?

The fact that the NRs form only via a complicated kinetic process involving Si/Ag exchange strongly indicates that the NRs and the Ag surface reconstruction are interdependent. As noted above, the authors compare NR adsorption energies and NR cohesive energies in order to claim that the NRs might be decoupled from the surface. This argument is not completely convincing because the adsorption energy calculation uses the energy of the free Si atom (Eq. 1) and of the MR-reconstructed Ag surface. However, they themselves show that the NR formation is seeded by clusters already formed on the surface, and that Si/Ag exchange must occur in some kinetic process during NR growth. Thus the Si "reservoir" is most unlikely to be that of the free atom. Such aspects are missing from this simple treatment: the "more correct approach", as they say on page 5, is the thermodynamic treatment encapsulated in Eqs.2 and 3, and which yields the (very strong) evidence for the correctness of the P-MR model.

My second main comment regards their usage of the term "penta-silicene". To quote from their rebuttal:

> It seems that the referee keeps on assuming that by "penta-silicene" we mean "silicene", but there is no scientific reason for doing so.

This is a highly telling statement. It is fine to talk about pentagons or pentagon chains (shapes), and acceptable to talk of pentamer or pentamer chains (adsorbate geometry). However, "silicene" refers to both geometry and electronic structure - this is, after all, why silicene is such a hot topic, and my feeling is that the authors are insistent on using this name because of its appeal, in spite of there being "no scientific reason for doing so", and "We make no claim (or even suggest) that the NRs share similar properties with those of silicene". Nonetheless, keeping the word "silicene" (even as "penta-silicene") is obfuscating, and make the system sound more interesting than it is (overselling). I cannot, as requested, propose a more "appealing" name, but more "appropriate" names include: pentagonal SiNRs/pentamer ribbons/fused pentamer rows/chains, and so on. Last,

I remind anyway that only integer multiples of single NRs are observed, casting doubt on possible formation of a penta-silicene sheet, which was anyway shown by Ding et al [9] to be dynamically unstable.

To summarize: As stated also in my original report, I do believe the authors have solved the mystery of the Si/Ag(110) structure, although confirmation by other methods will give the definitive result. I would like to stress that the work itself I regard to be of very high quality. On the other hand, I do not think this finding is, in itself, remarkable enough for Nature Communications, nor have the authors demonstrated, beyond some speculative comments, why it may become remarkable.

Reviewer #3 (Remarks to the Author):

In this improved manuscript, the authors claim again the originality and novelty of SiNRs structural model with pentagonal rings and missing rows on Ag(110) substrate. The simulations are convincing and corresponding experimental evidence is impressive. I keep my opinion that it is reasonable and can be published on Nature communications.

Further two comments:

- 1.To avoid misunderstanding, I suggest not to use "silicene", which usually means two-dimensional graphene-like Si phase. If possible, "Penta-SiNRs" is much better than the arguable "penta-silicene".
- 2.Under current situation, the authors did not give the mechanism clearly from the possible "hexagonal precursor" to pentagonal SiNRs. Then, the discussion for Fig. 1(a) is not significant or even improper.

Reviewer #2 (Remarks to the Author):

In the resubmitted version of the manuscript, the authors have addressed quite well many of the reviewers' issues. (For clarity, I am the original Reviewer #2.)

In response to Reviewer #1, they have attempted to explain the hexagonal-pentagonal transition from seed to nanoribbon (NR), unfortunately without success, and report detailed results in the Supplementary Material.

This is interesting work and a nice addition. They have also tested the importance of van der Waals corrections, finding them negligible (incidentally, this finding was also reported for other NR models in Ref. 22), in support of their total energy calculations. Last, they report Fermi velocities for some of their computed bands, which compare well with the experimental values. While this is further support of their model, one must be careful not to infer too much about the electronic nature of these bands, which can arise from adsorbate-induced umklapp processes, as shown in Gori et al, J. Appl. Phys. 114, 113710 (2013).

In response to Reviewer #3, the authors present an example of a Si/Ag exchange process on the (110) surface, showing that such a process is energetically viable (a finding reported in Ref. 22, now referred to correctly), and comment on the observed NR dislocations. They also discuss the new Supp. Fig. 8, i.e. stability of isolated NRs versus the adsorbed system, and report that the average adsorption energy is dominated by the internal cohesive energy.

Although this is a simplification (see below), it is an indication that the NRs might be detached from the surface.

At this point I feel that the authors have thoroughly addressed the issues raised by Reviewers 1 and 3.

We are glad that the referee finds satisfactory our replies and the changes made after reviewers #1 and #3 remarks as well as the fact that the new Supplementary Figure 8 is "interesting and a nice addition." Regarding the irrelevance of van der Waals corrections, we have added a sentence in the Methods section in order to acknowledge the similar conclusion reached in Ref. [22].

We also agree with the referee that we prefer to remain cautious when comparing the theoretical and experimental band structures, and that is why we do not mention in the manuscript the experimental Fermi velocity extracted from Fig. 12 in Ref.[28] (although it was included in our reply to Referee #1).

Considering now the response to Reviewer #2 comments (mine). My initial report was very long and detailed, and I appreciate that the authors have considered each point, sometimes agreeing and sometimes rebutting them. They offer some new calculations in the supplementary material but have not made any major changes or additions to the body of the paper. Below I will try to condense the review into two points, which should make my judgement also more clear to the editors.

First (the easier point): have the authors adequately shown that their pentagonal (P-MR) structural model is correct?

In my opinion: yes. In brief, I accept their arguments that the (independent) coverage measurements may be systematically wrong; that the STM images are well explained by the P-MR structure; that the 2:1 ratio from PES is a valid measurement now that the

errors have been explicitly reported; and that the evidence presented is adequate for publication. I accept that more experimental analysis is beyond the authors current scope and timeframe. The new calculations studying the hexagon-pentagon transition are a welcome addition, even though the matter remains unexplained. References to previous works have been improved, making the novel aspect of the present work more clear to the reader. Overall, therefore I am quite satisfied with the main content of the paper from page 1 (line 34) to page 3 (line 147): this text, which constitutes the bulk of the paper, puts forward a strong case for solving the structure of the Si NRs on Ag(110). The methodology is sound and the supplementary material is supportive, although it does not push the paper to a higher level of impact.

We sincerely acknowledge the reviewer for accepting our arguments, which unambiguously point towards the correctness of the pentagonal model.

Second: are their findings worthy of publication in Nature Communications?

In my opinion: no. I believe that the authors have identified and solved, to quote again from Ref 26: "a Ag(110) reconstruction induced by Si", but fail to demonstrate its general importance. In their rebuttal they claim that many of my objections are "subjective", but frankly, the onus is on them to show that their model--certainly aesthetically pleasing and, in a limited sense, unique--is worthy of publication in such a wide interest, high impact journal. The justification put forth is mostly contained in the abstract (page 1) and the last six lines of the main text.

Their most important claims in the manuscript are

- the work "puts hope on the realization of a new 1D Si allotrope" and (in the rebuttal) that "we hope our current work will motivate works aiming to realise quasi-free standing Si pentagonal structures"

If the authors succeeded in realizing a new Si allotrope, then, as they say in their rebuttal, they would be looking at a major scientific discovery. Without this, I fail to see anything remarkable about the system, except that the silicon therein is "solely comprising pentagonal rings". As the authors were not impressed by my citing of Si pentamer (isolated pentagons) formation on vicinal Si surfaces, I recall again the well studied case of In chains on Si(111). In that system, In forms chains of trimers or hexagons, and the system undergoes interesting phase transitions between the two. Here "phase" refers to the Si-In ensemble, not to the In chain structure. But I have not heard of any claims of "novel hexagonal In allotropes". It is simply not correct to consider only the silicon part of the reconstruction and claim it is a "silicon phase". There are many such adsorbate-stabilized 1D and 2D surface reconstructions: some reveal remarkable physical properties, and some do not. What is the remarkable physical property here?

We do find remarkable that this is the first Si-based structure "solely comprised of pentagonal rings", and this is certainly the main claim of the paper. The possibilities that this discovery opens are manifold, including Si-based nano-wires/circuits, enlarged spin-orbit effects, an unprecedented 1D topography or the realization of a new Si allotrope. Any of these exciting ideas fall beyond the scope of the current work, which is devoted to the determination and characterization of this Si-pentagonal structure stabilized on Ag(110) which, considering the long period it has remained elusive, is not at all a trivial task.

On the other hand, we again recall that the complex Si reconstructions pointed out by

the referee comprising pentamer motifs (among many other Si polygons) are just not comparable with the pentagonal nano-ribbons here determined given the simplicity and “pentagonality” of the latter (we believe this is perfectly clear after inspection of **Figure RR5** in our previous reply). The referee now invokes an interesting In reconstruction on Si(111), but it is clear that in such system, and because of the metallic nature of the In-In bonds, detaching the In chains from the substrate while retaining their hexagonal/trimer topography is completely hopeless. The scenario is completely different in the present case, as the structure involves strong covalent Si-Si bonds and, hence, a "clean" detachment of the chains seems more plausible. In this sense, we do not understand why the referee cannot envisage an analogy between our system and, for example, that of graphene chemisorbed on **different metal surfaces**.

The fact that the NRs form only via a complicated kinetic process involving Si/Ag exchange strongly indicates that the NRs and the Ag surface reconstruction are interdependent. As noted above, the authors compare NR adsorption energies and NR cohesive energies in order to claim that the NRs might be decoupled from the surface. This argument is not completely convincing because the adsorption energy calculation uses the energy of the free Si atom (Eq. 1) and of the MR-reconstructed Ag surface. However, they themselves show that the NR formation is seeded by clusters already formed on the surface, and that Si/Ag exchange must occur in some kinetic process during NR growth. Thus the Si "reservoir" is most unlikely to be that of the free atom. Such aspects are missing from this simple treatment: the "more correct approach", as they say on page 5, is the thermodynamic treatment encapsulated in Eqs.2 and 3, and which yields the (very strong) evidence for the correctness of the P-MR model.

We find hard to follow the referee's objection here. Obviously, the pentagonal chains are not spontaneously created, but require the Ag(110) surface which acts as the template (with the creation of missing rows) on which the strands form. But this constitutes the nano-ribbon (NR) formation stage, after which further manipulation may be attempted on these almost perfect extended ribbons. In fact, equation (1) is not necessary to assess some relevant energetic arguments. One may alternatively evaluate the NR-Ag(110) interaction, $E_I(\text{NR}, \text{Ag}_{110})$, via:

$$E_I(\text{NR}, \text{Ag}_{110}) = -E(\text{N}_{\text{Ag}}, \text{N}_{\text{Si}}) + E_{\text{surf}}(\text{N}_{\text{Ag}}) + E_{\text{NR}}^0,$$

where E_{NR}^0 is the total energy of the free-standing Si pentagonal NR. The resulting energy gain upon adsorption is $E_I(\text{NR}, \text{Ag}_{110}) = 1.5 \text{ eV/Si}$.

Similarly, the strength of the Si-H bonds in the hydrogenated free-standing ribbons (NRH) may be obtained from:

$$E_I(\text{NR}, \text{N}_{\text{H}}) = -E(\text{N}_{\text{Si}}, \text{N}_{\text{H}}) + E_{\text{NR}}^0 + N_{\text{H}} E_{\text{H}}^0,$$

where $E(\text{NR}, \text{N}_{\text{H}})$ is the total energy of the NRH with N_{H} hydrogens and E_{H}^0 that of an isolated hydrogen atom. For the three NRH structures considered in the Supplementary Figure 8, $N_{\text{H}} = 2, 4$ and 6 , we find that this energy gain is $E_I(\text{NR}, \text{N}_{\text{H}}) = 1.4, 2.9$ and 4.1 eV/Si , respectively, which implies that the NR would prefer to be hydrogenated (with 4 or 6 hydrogens) rather than adsorbed on the Ag(110) surface. Such simple count merely indicates that detaching the NR from the substrate via its hydrogenation would lead to an energetically more favorable situation and is, therefore, at least plausible. The actual mechanism of the hypothetical detachment and hydrogenation remains to be explored and, of course, the thermodynamic balance will depend on the H chemical potential as well as many other factors related to the experimental conditions which are beyond the scope of the current work.

Summarizing, we agree with the referee that we cannot assess that the NRs can be decoupled, but we cannot either find any argument proving that they cannot.

My second main comment regards their usage of the term "penta-silicene". To quote from their rebuttal:

It seems that the referee keeps on assuming that by "penta-silicene" we mean "silicene", but there is no scientific reason for doing so.

This is a highly telling statement. It is fine to talk about pentagons or pentagon chains (shapes), and acceptable to talk of pentamer or pentamer chains (adsorbate geometry). However, "silicene" refers to both geometry and electronic structure - this is, after all, why silicene is such a hot topic, and my feeling is that the authors are insistent on using this name because of its appeal, in spite of there being "no scientific reason for doing so", and "We make no claim (or even suggest) that the NRs share similar properties with those of silicene". Nonetheless, keeping the word "silicene" (even as "penta-silicene") is obfuscating, and make the system sound more interesting than it is (overselling). I cannot, as requested, propose a more "appealing" name, but more "appropriate" names include: pentagonal SiNRs/pentamer ribbons/fused pentamer rows/chains, and so on. Last, I remind anyway that only integer multiples of single NRs are observed, casting doubt on possible formation of a penta-silicene sheet, which was anyway shown by Ding et al [9] to be dynamically unstable.

Since we have no intention to oversell our results or confuse the reader and given that Reviewer #3 also poses concerns on the term “penta-silicene”, we have replaced it by “Si pentagonal nano-ribbons” everywhere in the text and the title.

To summarize: As stated also in my original report, I do believe the authors have solved the mystery of the Si/Ag(110) structure, although confirmation by other methods will give the definitive result. I would like to stress that the work itself I regard to be of very high quality. On the other hand, I do not think this finding is, in itself, remarkable enough for Nature Communications, nor have the authors demonstrated, beyond some speculative comments, why it may become remarkable.

We acknowledge again the reviewer for praising the “very high quality” of our work. However, as stated in our replies above, we disagree with him/her on its relevance, which we strongly feel is sufficiently high for Nature Communications.

Reviewer #3 (Remarks to the Author):

In this improved manuscript, the authors claim again the originality and novelty of SiNRs structural model with pentagonal rings and missing rows on Ag(110) substrate. The simulations are convincing and corresponding experimental evidence is impressive. I keep my opinion that it is reasonable and can be published on Nature communications. Further two comments:

1.To avoid misunderstanding, I suggest not to use “silicene”, which usually means two-dimensional graphene-like Si phase. If possible, “Penta-SiNRs” is much better than the arguable “penta-silicene”.

As the referee suggests, as well as Reviewer #2, we have changed the term “penta-silicene” by “Si pentagonal nano-ribbons” throughout the entire text and the title.

2.Under current situation, the authors did not give the mechanism clearly from the possible “hexagonal precursor” to pentagonal SiNRs. Then, the discussion for Fig. 1(a) is not significant or even improper.

We understand the reviewer’s concern regarding the appearance of the quasi-hexagonal nano-dot structure in Figure 1(a), but we would prefer to keep it since we believe it still holds key ingredients to understand the formation of the P-MR NRs which, as stated in the text, are: (i) the removal of top silver atoms and (ii) the presence of two types of Si atoms in the created vacancies. Indeed, the determination of the nano-dot atomic structure turned out to be essential for us in order to reach our final pentagonal structure. Furthermore, reviewers #1 and #2 seem satisfied with the analysis presented in the Supplementary Figure 7 which obviously requires the presentation of the nano-dot figure at some point in the manuscript.